# Sequential Least-Squares Estimators with Fast Randomized Sketching for Linear Statistical Models

## Abstract

We propose a novel randomized framework for the estimation problem of large-scale linear statistical models, namely Sequential Least-Squares Estimators with Fast Randomized Sketching (SLSE-FRS), which integrates *Sketch-and-Solve* and *Iterative-Sketching* methods for the first time. By iteratively constructing and solving sketched least-squares (LS) subproblems with increasing sketch sizes to achieve better precisions, SLSE-FRS gradually refines the estimators of the true parameter vector, ultimately producing high-precision estimators. We analyze the convergence properties of SLSE-FRS, and provide its efficient implementation. Numerical experiments show that SLSE-FRS outperforms the state-of-the-art methods, namely the Preconditioned Conjugate Gradient (PCG) method, and the Iterative Double Sketching (IDS) method.

## 1 Introduction

Linear regression is a fundamental model in both statistics and machine learning, widely used to capture relationships between variables. Suppose that there exists a standard linear relationship between the response vector $Y \in \mathbb{R}^N$ and the feature matrix $X \in \mathbb{R}^{N \times d}$ with the sample size $N$ and the feature size $d$ as follows

$$Y = X\beta + \zeta,$$

where $\beta \in \mathbb{R}^d$ is the unknown true parameter vector to be estimated and $\zeta \in \mathbb{R}^N$ represents the random noise vector with zero mean and a variance matrix $\sigma^2 I_N$.

To learn the parameter $\beta$, we consider the ordinary least-squares (OLS) estimator $\hat{\beta}$,

$$\hat{\beta} = \arg\min_{\beta \in \mathbb{R}^d} f(\beta; X, Y), \tag{1}$$

with $f(\beta; X, Y) = \frac{1}{2}\|Y - X\beta\|_2^2$. Throughout the paper, we assume that $X$ has full column rank, then the OLS estimator can be explicitly formulated as

$$\hat{\beta} = (X^\top X)^{-1} X^\top Y.$$

Due to its well-established and favorable statistical properties (Chatterjee & Hadi, 2009), the OLS estimator has been extensively employed to estimate $\beta$ in practice. However, for large-scale problems with $N \gg d$, the direct computational complexity $O(Nd^2)$ to obtain $\hat{\beta}$ becomes prohibitive. To address this challenge, numerous randomized algorithms based on *sketching* methods have been developed to approximate the OLS estimator efficiently.

The first classical randomized approach to reduce the computational cost, known as *Sketch-and-Solve* (Drineas et al., 2011; Sarlos, 2006), is using *sketching* matrix $S \in \mathbb{R}^{m \times N}$ with $m \ll N$ to construct the *sketched data* $(SX, SY)$. One can solve the following *smaller* sketched LS problem to obtain the sketched LS estimator $\tilde{\beta}$ as an approximation,

$$\tilde{\beta} = \arg\min_{\beta \in \mathbb{R}^d} f(\beta; SX, SY),$$

with $f(\beta; SX, SY) = \frac{1}{2}\|SY - SX\beta\|_2^2$. Then the direct methods can be called to compute the sketched estimator $\tilde{\beta}$ within $O(md^2)$ time. As the *suboptimality* illustrated in (Pilanci & Wainwright, 2016), any *Sketch-and-Solve* methods based on only observing a single pair of sketched

data $(SX, SY)$, unless the sketch size $m \geq N$, necessarily has a substantially larger error than the OLS estimator. In other words, with a small sketch size $m$, the *Sketch-and-Solve* methods result in estimators with relatively low precision.

The second widely adopted approach is *Iterative Sketching*, which involves repeatedly sketching the problem and iteratively refining the estimator. One popular method, Iterative Hessian Sketch (IHS), (Pilanci & Wainwright, 2016) uses a refreshed sketched Hessian matrix $H_t = X^\top S_t^\top S_t X$ to approximate the Hessian matrix $H = X^\top X$ of (1). The update formula can be expressed as:

$$\beta_{t+1} = \beta_t - H_t^{-1} \nabla f(\beta_t; X, Y),$$

where the sketching matrices $S_0, \ldots, S_t, \ldots$ are independent and identically distributed (i.i.d.) of size $m \times N$, with $m \ll N$ and $\nabla f(\beta; X, Y) := X^\top(X\beta - Y)$. In (Ozaslan et al., 2019), the convergence rate of IHS is significantly improved by incorporating a momentum term. See more related work in (Pilanci & Wainwright, 2017; Tang et al., 2017; Lacotte et al., 2021; Lacotte & Pilanci, 2020; 2021; Na et al., 2023; Epperly, 2024)

Inspired by the results in (Dobriban & Liu, 2019), we noticed that the asymptotic precision of the sketched LS estimator of the *Sketch-and-Solve* method can be explicitly formulated by a function of the sketch size $m$, the sample size $N$ and the feature size $d$. For a fixed sketching matrix (e.g., Gaussian or SRHT), increasing $m$ improves the precision of the estimator but also raises the computational costs. This theory appears to suggest that we can only strike a balance between improving accuracy and reducing computational complexity. In fact, it provides us with an opportunity to enhance the precision while simultaneously decreasing the computational costs.

We suggest applying the *Sketch-and-Solve* method multiple times with a carefully constructed sequence of sketched LS subproblems with increasing sketch sizes. The precision of the estimators can be iteratively improved. Each sketched LS subproblem can be solved using any efficient iterative LS solver. Importantly, compared to directly applying the solver to the original problem, the cost of performing iterations in the sketched LS subproblems is significantly cheaper. If the solution of an appropriate precision for each subproblem can be obtained at a relatively low cost, this new idea will lead to a substantial reduction in the overall computational expense.

Therefore, we propose a novel framework, named Sequential Least-Squares Estimators with Fast Randomized Sketching (SLSE-FRS). SLSE-FRS repeatedly applies the *Sketch-and-Solve* method with increasing sketch size to compute the estimators for the unknown parameter vector $\beta$. To the best of our knowledge, this concept is proposed for the first time in this area. However, three key issues need to be addressed.

The first issue is how to construct the sequence of the sketched LS subproblems effectively and efficiently. Since the sketched LS subproblems are constructed using the sketched data $(S_i X, S_i Y)$ for $i = 1, 2, \ldots$, where $S_i \in \mathbb{R}^{m_i \times N}$ is the sketching matrix of the sketch size $m_i$. Constructing each sketched LS subproblem necessitates accessing the original data once, resulting in a computational cost of at least $O(Nd)$. If we independently construct them, this cost becomes unacceptable. Moreover, selecting an appropriate sketch size is crucial, as it directly impacts both the precision and the computational cost.

The second issue is how to determine the stopping criterion for each sketched LS subproblem. Due to Theorem 1 in (Pilanci & Wainwright, 2016), the error between the sketched LS estimator and the true parameter $\beta$ has a lower bound, limiting achievable precision. In each sketched subproblem, we only need to achieve this level of precision. In practice, $\beta$ is unobservable. This theoretical quantity is not an accessible stopping criterion for the iterations. Thus, it is necessary to develop a theoretically rigorous and computationally feasible surrogate to serve as the appropriate stopping condition for each sketched LS subproblem.

The final issue is to ensure theoretically and numerically that SLSE-FRS achieves the same level of precision as the OLS estimator (the noise level or $\sigma$-level).

In this paper, we present a detailed introduction to the SLSE-FRS framework, address the above three issues, and demonstrate its superior performance through theoretical analysis and numerical experiments. Throughout the paper, we denote by $\|\beta\| := \|\beta\|_2$ the Euclidean norm of a vector $\beta$, and $\|M\| := \|M\|_2$ the spectral norm of a matrix $M$. The expectation $\mathbb{E}[\cdot]$ is taken over the random noise $\zeta$.

### 1.1 ADDITIONAL RELATED WORK

Another popular approach for solving the problem (1) is sketch-and-precondition(Martinsson & Tropp, 2020), which uses sketching to construct a preconditioner and combines it with classical iterative algotithm in a minimal number of iterations. One of the most prominent sketch-and-precondition techniques is known as Blendenpik(Avron et al., 2010), which can be considerably faster than the LS solver implemented in LAPACK. In addition, (Lacotte & Pilanci, 2021) presents comparison of multiple randomized algorithms and demonstrates that PCG with a preconditioner based on the Subsampled Randomized Hadamard Transform (SRHT) achieves the best numerical performance. Besides above three approaches, the summaries of the classical randomized sketching methods can be found in (Woodruff et al., 2014; Drineas & Mahoney, 2016; Martinsson & Tropp, 2020; Dereziński & Mahoney, 2024) and references therein.

## 2 THE SLSE-FRS FRAMEWORK

In this section, we will introduce the general framework of SLSE-FRS, a new iterative method to repeatedly apply the *Sketch-and-Solve* methods to obtain the estimators for the true parameter vector $\beta$.

The inspiration comes from (Dobriban & Liu, 2019), which analyzed the limits of precision loss incurred by the popular *Sketch-and-Solve* methods. We consider one of the loss functions, named the *relative prediction efficiency* (PE).

$$\text{PE} = \frac{\mathbb{E}\|X\tilde{\beta} - X\beta\|^2}{\mathbb{E}\|X\hat{\beta} - X\beta\|^2}. \tag{2}$$

PE measures the precision loss between the sketched LS estimator $\tilde{\beta}$ and the OLS estimator $\hat{\beta}$ due to the sketching in the *Sketch-and-Solve* methods. Given data $X$, PE depends on the sketch size $m$, the sample size $N$, and the feature size $d$. PE decreases with an increasing sketch size $m$, yielding a higher-precision estimator. This naturally motivates the idea of constructing a sequence of sketched LS subproblems with increasing sketch sizes to compute estimators with progressively higher precisions.

Popular LS solvers iteratively solve the original problem (1). In contrast, we suggest to iteratively solve a sequence of relatively small-scale sketched LS subproblems. Our framework proceeds in two stages. The 1st stage involves constructing and iteratively solving the sketched LS subproblems to compute estimators with progressively higher precisions. The solution of one sketched LS subproblem is used as the initial guess for the next. If the iterates can follow a sufficiently accurate path toward the true parameter vector $\beta$, our method can greatly reduce the computational cost. In the 2nd stage, we solve the full-scale LS problem to refine the estimator to the OLS-level precision. Since we already have an approximate estimator with a reasonable level of precision at the 1st stage, the 2nd stage can be completed within a few iterations at a very low cost. The goal is to achieve the precision of the OLS estimator with the lowest computational cost.

The new framework can be regarded as an inner-outer iteration method, where solving the sequence of LS subproblems constitutes the outer iteration, and the iteration for each LS subproblem forms the inner iteration. In detail, we construct $K$ sketched LS subproblems,

$$\min_{\beta \in \mathbb{R}^d} \frac{1}{2}\|S_i X\beta - S_i Y\|^2, i = 1, \ldots, K, \tag{3}$$

where $S_i \in \mathbb{R}^{m_i \times n}$ is the $i$-th sketching matrix and $(\tilde{X}, \tilde{Y}) := (S_i X, S_i Y)$ is the sketched data. Assume that the matrix $\tilde{X}^\top \tilde{X}$ has full rank, we have the $i$-th sketched LS estimator

$$\tilde{\beta}^i = (\tilde{X}^\top \tilde{X})^{-1}\tilde{X}^\top \tilde{Y}. \tag{4}$$

The sketched LS estimator $\tilde{\beta}^i$ can be viewed as an approximate estimator for the true parameter vector $\beta$. Therefore, we consider achieving the same level of precision of $\tilde{\beta}^i$ when an iterative LS solver is applied to the sketched LS subproblem (3). By defining the mean squared prediction error of the $i$-th exact sketched LS estimator $\tilde{\beta}^i$ relative to $\beta$ as below

$$\delta_i := \mathbb{E}\|X(\tilde{\beta}^i - \beta)\|, \tag{5}$$

**Algorithm 1** The SLSE-FRS framework

**Input:** $X, Y, \{(S_i X, S_i Y)\}_{i=1}^K, \{a_i\}_{i=1}^K,$

$T, T^\dagger \leftarrow \sum_{i=1}^K a_i, t \leftarrow 0$

### 1ST STAGE ###
**for** $i \leftarrow 1$ **to** $K$ **do**
  **for** $j \leftarrow 1$ **to** $a_i$ **do**
    $\beta_{t+1} \leftarrow \text{LS\_Solver}(\beta_t, S_i X, S_i Y)$
    $t \leftarrow t + 1$
  **end for**
**end for**
### 2ND STAGE ###
**for** $t \leftarrow T^\dagger$ **to** $T$ **do**
  $\beta_{t+1} \leftarrow \text{LS\_Solver}(\beta_t, X, Y)$
**end for**
Return $\beta_T$

**Algorithm 2** An Efficient SLSE-FRS

**Input:** $X, Y, \hat{H}, (S_0 X, S_0 Y), \{a_i\}_{i=1}^K, K, T$

$T^\dagger \leftarrow \sum_{i=1}^K a_i, \mu, \eta, \{m_i\}_{i=1}^K, \{B_i\}_{i=1}^K, t \leftarrow 0$

### 1ST STAGE ###
**for** $i \leftarrow 1$ **to** $K$ **do**
  $(S_i X, S_i Y) \leftarrow \sqrt{\frac{N}{m_i}} B_i (S_0 X, S_0 Y)$
  **for** $j \leftarrow 1$ **to** $a_i$ **do**
    $\beta_{t+1} \leftarrow \beta_t - \mu \hat{H}^{-1} \nabla f(\beta_t; S_i X, S_i Y) + \eta(\beta_t - \beta_{t-1})$
    $t \leftarrow t + 1$
  **end for**
**end for**
### 2ND STAGE ###
**for** $t \leftarrow T^\dagger$ **to** $T$ **do**
  $\beta_{t+1} \leftarrow \beta_t - \mu \hat{H}^{-1} \nabla f(\beta_t; X, Y) + \eta(\beta_t - \beta_{t-1})$
**end for**
Return $\beta_T$

for the $i$-th LS subproblem, we hope the adopted iterative LS solver can return an estimator $\beta^i$ satisfying the following condition

$$\mathbb{E}\|X(\beta^i - \beta)\| < [1 + o(1)]\delta_i, \tag{6}$$

which matches the best precision that an estimator can attain in the $i$-th LS subproblem. At this point, we have completed the 1st stage, and obtained $K$ estimators $\{\beta^i\}_{i=1}^K$ with progressively higher precisions. Since we aim to achieve the precision of the OLS estimator, due to the suboptimality of the *Sketch-and-Solve* method, we move on to the 2nd stage, namely apply an iterative LS solver to the full-scale LS problem. Therefore, we utilize the estimator $\beta^K$ from the iterative solution of the $K$-th sketched LS subproblem as the initial guess, which is a high-quality estimator for $\beta$, only a few additional iterations are required to achieve the OLS precision, leading to significant cost savings.

In the SLSE-FRS framework, we only need to ensure that the iterative solution error of the $i$-th sketched LS subproblem reaches the order of $\delta_i$ for $i = 1, \ldots, K$. Therefore, SLSE-FRS is not limited to any specific LS iterative solver but is compatible with all efficient LS iterative solvers. This compatibility is very powerful as it allows for the seamless integration of any current and future efficient iterative LS solvers within SLSE-FRS. Additionally, compared to directly using LS iterative solvers on the original full-scale LS problem, SLSE-FRS significantly reduces computational costs by applying iterative LS solvers on small-scale sketched LS subproblems. We summarize the general SLSE-FRS framework in Algorithm 1, where $a_i$ represents the number of iterations performed for the $i$-th sketched LS subproblem to meet the stopping criterion, namely the condition (6).

Now, we consider the efficient construction of the $K$ sketched LS subproblems. The error $\delta_i$ should gradually approach the precision of the OLS estimator, which means that the sketch size should increase according to a specific pattern. We adopt a similar idea of obtaining the sequence of the sketched data in (Wang et al., 2022). Specifically, the data $(S_i X, S_i Y)$ can be easily extracted from the data $(S_{i+1} X, S_{i+1} Y)$, and the sketch sizes satisfy $m_2/m_1 = \cdots = m_K/m_{K-1}$.

Moreover, different LS solvers may adopt different stopping criteria when solving the LS subproblems. In Section 3, we will introduce an efficient implementation of the SLSE-FRS framework based on the Momentum-IHS (M-IHS) algorithm (Ozaslan et al., 2019), providing its stopping criteria and convergence analysis. This can serve as a reference for establishing relevant stopping criteria and convergence analysis when adopting other LS solvers for the LS subproblems.

## 3 AN EFFICIENT SLSE-FRS

After introducing the general framework of SLSE-FRS, we now present an efficient implementation. By considering the trade-off between sketching time and theoretical guarantee, we choose SRHT as sketching matrix to implement SLSE-FRS.

Here we construct the SRHT matrix in a similar fashion as (Dobriban & Liu, 2019). For an integer $N = 2^p$ with $p \geq 1$, We define the $N \times N$ SRHT matrix $S = (N/m)^{1/2} BHDP$, where $B$ is the $N \times N$ diagonal random sampling matrix with i.i.d. Bernoulli random variables with success probability $m/N$, $H$ is a normalized Hadamard matrix, $D \in \mathbb{R}^{N \times N}$ is a diagonal matrix with i.i.d. rademacher random variable, and $P \in \mathbb{R}^{N \times N}$ is a uniformly distributed permutation matrix. Lastly, We retain the non-zero rows of the matrix $S$, which forms the SRHT matrix, and still denote it as $S$. The following theorem implies the embedding property of the SRHT sketching matrix.

For the selection of the LS solver in the two stages of SLSE-FRS, we prefer to adopt the M-IHS algorithm due to its numerous beneficial features (Ozaslan et al., 2019). Moreover, the Hessian sketch applied in M-IHS is defined as $\hat{H} \triangleq (\hat{S}A)^T \hat{S}A$, where $\hat{S} \in \mathbb{R}^{r \times N}$ is a fixed sketching matrix.

In SLSE-FRS, the M-IHS iteration in the 1st stage can be expressed as

$$\beta_{t+1} = \beta_t - \mu \hat{H}^{-1}(S_i X)^\top (S_i X \beta - S_i Y) + \eta(\beta_t - \beta_{t-1}), \tag{7}$$

where $S_i$ is the $i$-th sketching matrix in the $i$-th sketched LS subproblem. The M-IHS iteration in the 2nd stage can be expressed as

$$\beta_{t+1} = \beta_t - \mu \hat{H}^{-1} X^\top (X\beta - Y) + \eta(\beta_t - \beta_{t-1}). \tag{8}$$

In fact, the formulae (7) and (8) can be regarded as using a number of preconditioned Richardson extrapolation iterations, namely $a_i$ iterations for (7) and $T - T^\dagger$ iterations for (8), with the preconditioner $\hat{H}$ to solve the normal equations of the sketched LS subproblems and the full-scale original LS problem, namely

$$(S_i X)^\top S_i X \beta = (S_i X)^\top S_i Y \tag{9}$$

and

$$X^\top X \beta = X^\top Y. \tag{10}$$

The floating-point operation (FLOPs) cost of the formulae (7) and (8) can be obtained based on the results in (Golub & Van Loan, 2013). Given $\beta_t$, $\hat{H}$, and the sketched data $(S_i X, S_i Y), i = 1, \ldots, K$, one sketched M-IHS iteration using formula (7) in the 1st stage requires $\{(4d+1)m_i + 2d^2 + 5d\}$ FLOPs, while one full-scale M-IHS iteration with the data $(X, Y)$ requires $\{(4d+1)N + 2d^2 + 5d\}$ FLOPs. In one iteration, the first term of the cost is reduced from $\{(4d+1)N\}$ to $\{(4d+1)m_i\}$. It is noteworthy that the number of full-scale M-IHS iterations required in the 2nd stage is significantly less than the number of full-scale M-IHS iterations used throughout the entire iteration process.

Based on the embedding property of SRHT, the SRHT sketching matrix $\hat{S} \in \mathbb{R}^{r \times N}$ adopted in the Hessian sketch $\hat{H} \triangleq (\hat{S}X)^T \hat{S}X$ should be of sketch size $r = O(d \log d)$. We follow the idea in (Wang et al., 2022) to determine the sketch size $m_i$, $i = 1, \ldots, K$, of the sketched LS subproblem (3), which grows by a factor of $2 = m_{i+1}/m_i$, $i = 1, \ldots, K-1$. In this way, we construct $K = \log_2(N/m_1)$ sketched LS subproblems. According to the aforementioned iterations for (9) and (10), the Hessian sketch $\hat{H}$ serves as a randomized preconditioner (or approximation) of both $(S_i X)^\top S_i X$ and $X^\top X$. Therefore, the sketch size of the smallest sketched LS subproblem should be larger than the sketch size of $\hat{H}$, i.e., $m_1 > r$.

Now, we consider the details of the efficient construction of the sketching matrix $S_i \in \mathbb{R}^{m_i \times N}$, $i = 1, \ldots, K$. Firstly, we left-multiply the original data $(X, Y)$ by $HDP$ to get the full-scale sketched data $(S_0 X, S_0 Y) := (HDPX, HDPY)$. Secondly, we can easily construct the sketched data sequence $(S_i X, S_i Y), i = 1, \ldots, K$, by sampling the rows of the data $(S_0 X, S_0 Y)$, namely by left-multiplying the random matrix $B_i \in \mathbb{R}^{m_i \times N}$, which is consisted of the $m_i$ non-zero rows of $B$. The $i$-th sketched data can be represented as

$$(S_i X, S_i Y) = \sqrt{\frac{N}{m_i}} B_i (S_0 X, S_0 Y).$$

The bottleneck of constructing the $K$ sketched LS subproblems is to apply the Hadamard transform to a matrix of size $N \times d$, which costs $Nd \log_2 N$ FLOPS. Hence, the overall sketching time complexity is $Nd \log_2 N$. In summary, the above discussion leads to an efficient implementation of SLSE-FRS, as described in Algorithm 2.

# 4 THE CONVERGENCE OF SLSE-FRS

This section offers the theoretical assurance of SLSE-FRS. Let $\beta_j^i$ represent the $j$-th iterate of the $i$-th sketched LS subproblem. The $K$ iterates $\beta_{a_i}^i$, $i = 1, \ldots, K$, of the sketched LS subproblems can serve as a sequence of estimators for the true parameter vector $\beta$ with increasing precision $\delta_i$. For any $\beta_0 \in \mathbb{R}^d$, we set $\beta_0^1 = \beta_0$ and $\beta_0^i = \beta_{a_{i-1}}^{i-1}, i = 2, \ldots, K$. The following theorem provides the convergence behavior of the iterations for the $i$-th sketched LS subproblem.

**Theorem 4.1.** *In the $i$-th sketched LS subproblem of SLSE-FRS in Algorithm 2, let $N, m_1$ be powers of 2, $\delta \in (0, 1)$, $\epsilon \in (0, 1/10)$, $|\mu - 1| \le 1/4$, and $\eta = 53/36 - \sqrt{17}/3$. Let*

$$r \ge c\epsilon^{-2} \left[ d + \log\left(\frac{N}{\delta}\right) \right] \log\left(\frac{ed}{\delta}\right), \quad m_1 > r,$$

*where $c > 0$ is a constant. There exists a constant $M_i > 0$, such that if $a_i > M_i$, with probability at least $1 - 2\delta$, we have*

$$\mathbb{E}\|X(\beta_{a_i}^i - \beta)\| \le (\frac{1}{3})^{a_i} \mathbb{E}\|X(\beta_0^i - \beta)\| + \left[1 + (\frac{1}{3})^{a_i}\right] \delta_i.$$

According to Theorem 4.1, the prediction error decays exponentially. For sufficiently large $a_i$, the estimator $\beta_{a_i}^i$ of the $i$-th sketched LS subproblem can achieve the precision $\delta_i$. Based on Theorem 4.1, we provide the convergence assurance of SLSE-FRS in the following theorem.

**Theorem 4.2.** *In Algorithm 2, let $N, m_1$ be powers of 2, $T^\dagger = \sum_{i=1}^K a_i$, $\delta \in (0, 1)$, $\epsilon \in (0, 1/10)$, $|\mu - 1| \le 1/4$, and $\eta = 53/36 - \sqrt{17}/3$. Let*

$$r \ge c\epsilon^{-2} \left[ d + \log\left(\frac{N}{\delta}\right) \right] \log\left(\frac{ed}{\delta}\right), \quad m_1 > r,$$

*where $c > 0$ is a constant. There exists a constant $M > 0$, such that if $a_i > M$, $i = 1, \ldots, K$, and $T > T^\dagger + M$, with probability at least $1 - 2\delta$, for any $\beta_0 \in \mathbb{R}^d$, it holds that*

$$\mathbb{E}\|X(\beta_T - \beta)\| \le (\frac{1}{3})^T \mathbb{E}\|X(\beta_0 - \beta)\| + [1 + o(1)]\mathbb{E}\|X(\hat{\beta} - \beta)\|, \tag{11}$$

*and $o(1)$ is an infinitesimal as $M \to +\infty$*

Under the parameter settings specified in Theorems 4.1 and 4.2, the convergence rate of SLSE-FRS is bounded above by $1/3$. We note that, according to the idea of the Proof of Theorem 4.1 in Appendix A.2, there is a possibility to obtain an even sharper upper bound for the convergence rate of SLSE-FRS by carefully selecting alternative parameter settings.

In practical implementations, the determination of the iteration count $a_i$ in the $i$-th sketched LS subproblem is essential. Here, we provide an estimation of $a_i$ required by the $i$-th sketched LS subproblem to achieve the precision $\delta_i$. In the $i$-th sketched LS subproblem, we expect that SLSE-FRS returns an iterate $\beta_{a_i}^i$ satisfying the condition (6). With a prescribed tolerance $\omega \in (0, 1)$, the count $a_i$ can be determined by requiring

$$\mathbb{E}\|X(\beta_{a_i}^i - \beta)\| \le \mathbb{E}\|X(\beta_{a_i}^i - \tilde{\beta}^i)\| + \mathbb{E}\|X(\tilde{\beta}^i - \beta)\|$$
$$\le (1 + \omega)\mathbb{E}\|X(\tilde{\beta}^i - \beta)\|. \tag{12}$$

The following theorem provides a lower bound of $a_i$, $i = 2, \ldots, K$. The lower bound of $a_1$ is determined by Theorem A.11 in Appendix A.4.

**Theorem 4.3.** *In Algorithm 2, for a prescribed tolerance $\omega \in (0, 1)$, the iteration count $a_i$ needed for the $i$-th sketched LS subproblem to full fill (12) satisfies*

$$a_i \ge \log_3 \left[ \frac{(1 + \omega)r(i - 1, i) + 1}{\omega} \right] > 0,$$

*where $r(i - 1, i) = \sqrt{m_{i-1} - d/m_i - d}$.*

Although the iteration count, e.g., $a_i$ and $T$, should be a positive integer, for convenience of explanation, we will directly use positive real numbers to represent the iteration count in the following discussion. If we let the iteration count $a_i$ equals to the above lower bounds, then Theorem 4.3 implies

that the sequence $\{a_i\}_{i=2}^K$ are monotonically increasing with respect to $i$, namely $a_2 < \ldots < a_K$ (refer to Appendix A.6), which can serve as a guidance for determining $a_i$ in Algorithm 2.

To end this section, we summarize the computational complexity of Algorithm 2 in the asymptotic sense in the theorem as follows. Here, the complexity is only considered for its main part, i.e., the dominant portion.

**Theorem 4.4.** *Under the same conditions as Theorem 4.3, let Algorithm 2 terminate when it finds an estimator that achieves the noise level $\sigma$. Let the positive integer $K = O(1)$, and the sketch sizes satisfy $m_{i+1}/m_i = 2$, $i = 1, \ldots, K - 1$, and $m_K = N/2$. For a specified $\omega \in (0, 1)$, let $a_i$ $(i = 1, \ldots, K)$ take exactly their lower bounds. In the sense of asymptotic meaning, that is, let $d/N \to \gamma \in (0, 2^{-K})$ as $N \to +\infty$, it holds that $\log_3(1/\omega) < a_i < \alpha$ $(i = 1, \ldots, K)$ with $\alpha = \log_3\{[1 + (1 + \omega)/\sqrt{2}]/\omega\}$ if $1 < \mathbb{E}\|X(\beta_0 - \tilde{\beta}_1)\|/\mathbb{E}\|X(\tilde{\beta}_1 - \beta)\| < 1 + (1 + \omega)/\sqrt{2}$. Hence, the costs (dominant portion only) of all stages of Algorithm 2 are listed below:*

- *The initialization stage: $Nd\log_2 N$;*

- *The 1st stage: $4\alpha Nd$;*

- *The 2nd stage: $4[\log_3(\omega^K/\sigma)]Nd$ with $\omega > \sigma^{1/K}$.*

According to Theorem 4.4, the complexity of Algorithm 2 is dominated by the initialization stage, namely $Nd\log_2 N$. On one hand, in scenarios where $\log_2 N$ can be considered as $O(1)$, the total complexity of Algorithm 2 is $O(Nd)$. On the other hand, if the SRHT sketching matrix is replaced by the CountSketch sketching matrix, the complexity at the initialization stage of Algorithm 2 can be reduced to $O(Nd)$. If theorems (currently non-existent) similar to Theorems 4.2-4.4 can be proved, the theoretical complexity of Algorithm 2 with CountSketch becomes $O(Nd)$. The experiments in the next section show that Algorithm 2 with CountSketch is faster than Algorithm 2 with SRHT in the sense of computing time.

## 5 NUMERICAL EXPERIMENTS

In this section, we present numerical experiments to show the effectiveness and efficiency of SLSE-FRS. We compare its performance with IDS in (Wang et al., 2022), PCG in (Lacotte & Pilanci, 2021), and M-IHS in (Ozaslan et al., 2019). Our experiments follow the standard configurations in prior works (Pilanci & Wainwright, 2016; 2017; Wang et al., 2022). See the environment details in Appendix A.10.

We begin with generating linear models $Y = X\beta + \zeta$. Based on these models, we construct LS test problems. The feature matrix $X$ is constructed by artificially adjusting the condition number of a matrix of size $N \times d$, which has i.i.d. entries distributed according to a Gaussian distribution, and the i.i.d. entries of the true parameter vector $\beta$ are also drawn from a Gaussian distribution. To evaluate the precision of each iterate $\beta_t$ for $t = 1, \ldots, T$, we consider the prediction error $\Delta_t := \|X(\beta_t - \beta)\|^2$. According to (Pilanci & Wainwright, 2016), the iterates are expected to achieve the LS error, defined as $\Delta := \|X(\hat{\beta} - \beta)\|^2$, which is approximately equal to $\sigma^2 d$. The elements of the noise vector $\zeta$ are i.i.d. sampled from $\mathcal{N}(0, 10^{-8})$. We then form the response vector $Y$ via the linear model.

The estimators of the true parameter vector $\beta$ are computed by SLSE-FRS, IDS, PCG, and M-IHS. We set $T = 100$ to ensure all algorithms converge and achieve the OLS precision (as small as the LS error $\Delta$) within this limit. In the following experiments, we test different sample sizes $N$, feature sizes $d$, condition numbers $\kappa$ of the feature matrix $X$, and the Hessian sketch size is set to $r = 6d$ for all test models. PCG uses the SRHT sketching matrix to construct the preconditioner like (Lacotte & Pilanci, 2021). IDS employs the iteration parameter $\mu = \frac{(1-d/r)^2}{1+d/r}$ as suggested in (Wang et al., 2022). SLSE-FRS and M-IHS adopt the iteration parameters $\mu = (1 - \eta)^2$ and $\eta = d/r$ following (Ozaslan et al., 2019). SLSE-FRS takes the expansion ratio $m_{i+1}/m_i = 2$ with $m_1 = 8d$ and $m_K = N/2$ for sketch sizes. For the feasible values of $a_i$ in SLSE-FRS, we analyzed its relationship with $\omega$ and observed that $a_i = 2$ yields the minimum computing time. Furthermore, the total computing time at $a_i = 3$ is very close to that at $a_i = 2$. See the detailed analysis in Appendix A.7. Based on the observations, we set $a_i = 2$ or 3 for all $i$ in subsequent experiments. The results of the subsequent report are the average of 10 independent runs.

## 5.1 PRECISION AND EFFICIENCY

This part is designed to validate two key aspects: the first is that the output of SLSE-FRS can achieve the LS error $\Delta$, and the second is that SLSE-FRS improves the state-of-the-art computing time for high-precision LS estimators.

First, we evaluate above algorithms on LS problems with $N \in \{2^{17}, 2^{18}, 2^{19}, 2^{20}\}$, $d = 2^6$, and $\kappa = 10^4$. As recommended, we take $a_i = 2$ in SLSE-FRS. In the first column of Figure 1, the top plot illustrates the prediction error $\Delta_T$ of SLSE-FRS and the LS error $\Delta$ of the OLS estimator (approximately $\sigma^2 d = 6.4 \times 10^{-7}$ with the noise variance $\sigma^2 = 10^{-8}$). This plot confirms that SLSE-FRS has achieved the OLS precision. Additionally, we construct a two-dimensional LS problem with $N = 2^{20}$ and $d = 2$ to visualize the convergence trajectories of SLSE-FRS and IDS. For each method, 100 independent convergence paths are plotted. As shown in the bottom plot of Figure 1, SLSE-FRS paths are more concentrated than those of IDS. This phenomenon arises because SLSE-FRS can better utilize the sketched LS estimators as control points for the iteration paths, thereby determining a more accurate and stable search direction.

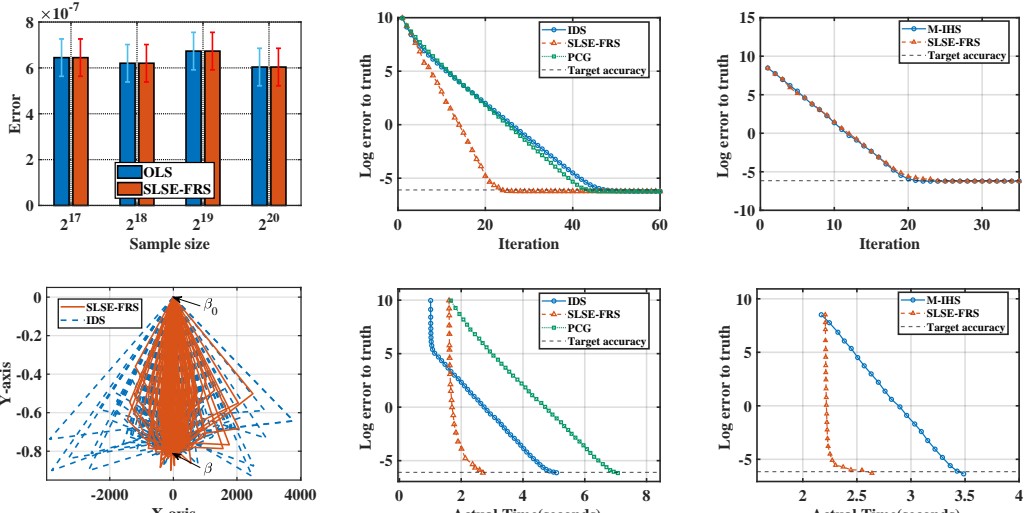

Figure 1: (i) First column: $\Delta_T$ of SLSE-FRS and $\Delta$ of OLS and convergence trajectories of SLSE-FRS and IDS; (ii) Second column: $\Delta_t$ versus iterations and actual computing time for SLSE-FRS, IDS, and PCG; (iii) Last column: $\Delta_t$ versus iterations and actual computing time for SLSE-FRS and M-IHS.

Next, we test the LS problem with $N = 2^{20}$, $d = 2^6$, and $\kappa = 10^4$. The second column of Figure 1 illustrates $\Delta_t$ versus iteration count and actual computing time in seconds of SLSE-FRS against IDS and PCG. As shown in the plots, SLSE-FRS converges much faster than PCG and IDS. Furthermore, SLSE-FRS demonstrates a significant improvement in computational efficiency. The computing time for IDS is approximately twice that of SLSE-FRS, while PCG takes roughly three.

We include M-IHS to further verify that SLSE-FRS achieves a convergence speed comparable to *Iterative-Sketching* methods directly applied to the full-scale LS problem while significantly reducing computational costs. Under the setting of $N = 2^{20}$, $d = 2^6$, $\kappa = 10^8$, the last column of Figure 1 clearly demonstrates the rationality of the SLSE-FRS framework design. SLSE-FRS strategically utilizes an optimal quantity of data in each iteration, ensuring efficiency and precision. In contrast, M-IHS may involve an excessive amount of data in early iterations, unnecessarily increasing computational costs.

Moreover, we provide one larger-scale experiment with $N = 2^{20}$, $d = 2^{10}$, $\kappa = 10^8$ in Appendix A.8. SLSE-FRS still maintains its superior performance in this setting.

Table 1 presents the computing time required for SLSE-FRS and the state-of-the-art methods (i.e., IDS and PCG) to achieve the target precision under different settings of $N \in \{2^{17}, 2^{18}, 2^{19}, 2^{20}, 2^{22}\}$, $d = 2^6$, and $\kappa \in \{10^4, 10^8\}$. As shown in this table, SLSE-FRS outperforms IDS and PCG. The computing time of IDS is twice that of SLSE-FRS, while the computing time of PCG is three times that of SLSE-FRS.

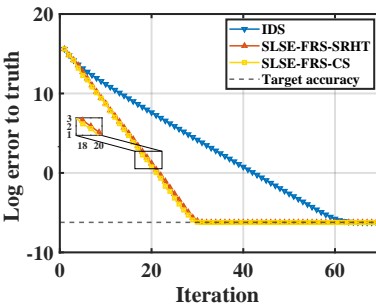 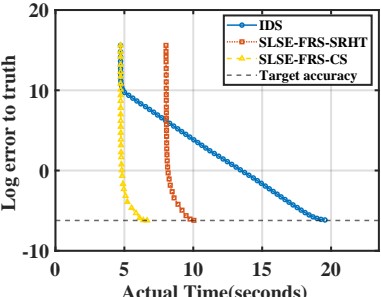

Figure 2: $\Delta_t$ versus iterations and actual computing time for SLSE-FRS-SRHT, SLSE-FRS-CS and IDS.

## 5.2 AN ALTERNATIVE SKETCHING MATRIX

It is noteworthy that the data sketching overhead of SLSE-FRS is relatively high, and the cost can be reduced by using a more computationally efficient sketching matrix. With the test problem under the setting of $N = 2^{22}$, $d = 2^6$, and $\kappa = 10^8$, we replace SRHT with CountSketch as the sketching matrix. In Figure 2, we have observed a similar convergence rate for both SLSE-FRS with SRHT (SLSE-FRS-SRHT) and SLSE-FRS with CountSketch (SLSE-FRS-CS), while the initialization time of SLSE-FRS-CS is significantly reduced. This experiment indicates that SLSE-FRS, combined with a computationally more efficient sketching matrix, has great potential to further improve its computational efficiency.

## 6 DISCUSSION

In this work, we introduced the novel SLSE-FRS framework for large-scale estimation problem of linear statistical models which iteratively constructs and solves sketched LS subproblems with increasing sketch sizes. It is the first algorithmic framework to integrate *Sketch-and-Solve* and *Iterative-Sketching* and can flexibly adapt to other potential integration schemes. We investigated the theoretical properties of SLSE-FRS. Experiments demonstrate that SLSE-FRS significantly improves the performance of the state-of-the-art methods. However, several challenges remain open for further investigation.

Although the LS solver M-IHS was used to construct an efficient SLSE-FRS implementation, other alternative LS solvers can also be applied to this novel framework. In future work, more LS solvers can be explored to address a wider range of scenarios.

The appropriate determination of $a_i$ and $m_i$ have significant importance for improving the computational efficiency of SLSE-FRS. We explored the tuning of the sketch size and $a_i$ in Appendix A.9. Currently, our study on them is still limited, and they deserve further exploration in the future.

In Section 5.2, we presented the heuristic results of SLSE-FRS with the more efficient CountSketch sketching matrix. Given that most of the computing time is attributed to initialization, it is worthwhile to consider replacing SRHT with CountSketch or other efficient alternatives, and study the related theories of SLSE-FRS with these alternative sketching strategies.

Table 1: Actual computing time for SLSE-FRS, IDS and PCG.

| COMPUTING TIME | $d = 2^6$ AND $\kappa = 10^4$ | | | | $d = 2^6$ AND $\kappa = 10^8$ |
|---|---|---|---|---|---|
| $N$ | $2^{17}$ | $2^{18}$ | $2^{19}$ | $2^{20}$ | $2^{22}$ |
| IDS | 0.4947 | 0.9145 | 1.7736 | 5.1532 | 19.1928 |
| PCG | 0.8087 | 1.4261 | 3.0466 | 7.2336 | 28.2938 |
| SLSE-FRS | **0.2524** | **0.4817** | **1.1510** | **2.5348** | **9.3281** |

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

# A APPENDIX

## A.1 NOTATIONS

The notation used in theoretical analysis is defined as follows. We denote by $X = U_X D_X V_X^\top$ the reduced singular value decomposition of $X$, where $U_X$ is a column orthonormal matrix of size $N \times d$, $V_X$ is a orthogonal matrix of size $d \times d$, and $D_X$ is a diagonal matrix of size $d \times d$. The variance denoted by $\mathrm{Var}[\cdot]$ is taken over the random noise $\zeta$.

For $\epsilon \in (0,1)$, we define the "good" events

$$\mathcal{E}_\epsilon := \bigcap_{i=1}^{K} \left\{ \left\| U^\top S_i^\top S_i U - I_d \right\| \le \epsilon \right\}$$

and

$$\hat{\mathcal{E}}_\epsilon := \left\{ \left\| U^\top \hat{S}^\top \hat{S} U - I_d \right\| \le \epsilon \right\}.$$

## A.2 PROOF OF THEOREM 4.1

In this subsection, we start with some useful lemmas. After that, we present the proof of this theorem.

**Lemma A.1.** *(Wang et al., 2022) Let $S$ be a SRHT sketching matrix of size $m \times N$. Let $N$ be a power of 2, $U \in \mathbb{R}^{N \times d}$ be a column orthonormal matrix, and $\epsilon, \delta \in (0,1)$. If*

$$m \ge c\epsilon^{-2} \left( d + \log\left( \frac{N}{\delta} \right) \right) \log\left( \frac{ed}{\delta} \right),$$

*where $c > 0$ is a constant, it holds that*

$$\Pr\left\{ \left\| U^\top S^\top S U - I_d \right\| > \epsilon \right\} \le \delta.$$

**Lemma A.2.** *Suppose the conditions of Theorem 4.1 hold. Then for any column orthonormal matrix $U \in \mathbb{R}^{N \times d}$, we have*

$$\Pr\left\{ \bigcup_{i=1}^{K} \left\{ \left\| U^\top S_i^\top S_i U - I_d \right\| > \epsilon \right\} \right\} \le \delta,$$

*and*

$$\Pr\left\{ \left\| U^\top \hat{S}^\top \hat{S} U - I_d \right\| > \epsilon \right\} \le \delta.$$

***Proof.*** For a sufficiently large constant $c$, Lemma A.1 ensures the following condition

$$\Pr\left\{ \left\| U^\top S_i^\top S_i U - I_d \right\| > \epsilon \right\} \le \frac{\delta}{K}, \quad i = 1, \dots, K.$$

Since we have $K$ sketched LS subproblems, by applying the union bound to the above $K$ events, the first conclusion follows.

The construction of $\hat{S}$ is independent of $S_i$ for $i = 1, \dots, K$. Using a similar argument, the second conclusion also holds. □

**Lemma A.3.** *For any column orthonormal matrix $U \in \mathbb{R}^{N \times d}$, conditioned on the event $\mathcal{E}_\epsilon$, for $i = 1, \dots, K$, we have*

$$\|(U^\top S_i^\top S_i U)^{-1}\| \le \frac{1}{1-\epsilon},$$

*and conditioned on the event $\hat{\mathcal{E}}_\epsilon$, we have*

$$\|(U^\top \hat{S}^\top \hat{S} U)^{-1}\| \le \frac{1}{1-\epsilon}.$$

**Proof.** Conditioned on the event $\mathcal{E}_\epsilon$, for $i = 1, \ldots, K$, we have

$$\|(U^\top S_i^\top S_i U)^{-1}\| = \|(I_d + U^\top S_i^\top S_i U - I_d)^{-1}\| \leq \frac{1}{1 - \|U^\top S_i^\top S_i U - I_d\|} \leq \frac{1}{1 - \epsilon}.$$

The second inequality follows in a similar way. $\qquad\square$

**Lemma A.4.** *For any column orthonormal matrix $U \in \mathbb{R}^{N \times d}$, conditioned on the event $\hat{\mathcal{E}}_\epsilon$, we have*

$$\|I_d - \mu(U^\top \hat{S}^\top \hat{S} U)^{-1}\| \leq \frac{\epsilon + |\mu - 1|}{1 - \epsilon}.$$

**Proof.** Conditioned on the event $\hat{\mathcal{E}}_\epsilon$, according to Lemma A.3, we have $\|(U^\top \hat{S}^\top \hat{S} U)^{-1}\| \leq \frac{1}{1-\epsilon}$. Hence,

$$\begin{aligned}
\|I_d - \mu(U^\top \hat{S}^\top \hat{S} U)^{-1}\| &= \|(U^\top \hat{S}^\top \hat{S} U)^{-1}(U_A^\top \hat{S}^\top \hat{S} U - \mu I_d)\| \\
&\leq \|(U^\top \hat{S}^\top \hat{S} U)^{-1}\|\|U^\top \hat{S}^\top \hat{S} U - \mu I_d\| \\
&\leq \|(U^\top \hat{S}^\top \hat{S} U)^{-1}\| \left( \|U^\top \hat{S}^\top \hat{S} U - I_d\| + |\mu - 1| \right). \\
&\leq \frac{\epsilon + |\mu - 1|}{1 - \epsilon}.
\end{aligned}$$

$\square$

**Lemma A.5.** *For any column orthonormal matrix $U \in \mathbb{R}^{N \times d}$, conditioned on the event $\hat{\mathcal{E}}_\epsilon \cap \mathcal{E}_\epsilon$, for $i = 1, \ldots, K$, we have*

$$\|I_d - \mu(U^\top \hat{S}^\top \hat{S} U)^{-1} U^\top S_i^\top S_i U\| \leq \frac{(\mu + 1)\epsilon + |\mu - 1|}{1 - \epsilon}.$$

**Proof.** Conditioned on the event $\hat{\mathcal{E}}_\epsilon \cap \mathcal{E}_\epsilon$, according Lemma A.4, we have $\|I_d - \mu(U^\top \hat{S}^\top \hat{S} U)^{-1}\| \leq \frac{\epsilon + |\mu - 1|}{1 - \epsilon}$. Hence, for $i = 1, \ldots, K$, it follows that

$$\begin{aligned}
&\|I_d - \mu(U^\top \hat{S}^\top \hat{S} U)^{-1} U^\top S_i^\top S_i U\| \\
&= \|I_d - \mu(U^\top \hat{S}^\top \hat{S} U)^{-1} + \mu(U^\top \hat{S}^\top \hat{S} U)^{-1}(I_d - U^\top S_i^\top S_i U)\| \\
&\leq \|I_d - \mu(U^\top \hat{S}^\top \hat{S} U)^{-1}\| + \mu\|(U^\top \hat{S}^\top \hat{S} U)^{-1}\|\|I_d - U^\top S_i^\top S_i U\| \\
&\leq \frac{\epsilon + |\mu - 1|}{1 - \epsilon} + \mu\frac{\epsilon}{1 - \epsilon} \\
&= \frac{(\mu + 1)\epsilon + |\mu - 1|}{1 - \epsilon}.
\end{aligned}$$

$\square$

**Proof of Theorem 4.1.** We consider the $i$-th sketched LS subproblem. In the case $i = 1$, we set the initial iterates $\beta_{-1}^1 = \beta_0^1 = \beta_0$. In the case $i > 1$, we set the initial iterates $\beta_{-1}^i = \beta_0^i = \beta_{a_{i-1}}^{i-1}$ based on the iterates from the previous sketched LS subproblem. The update formula is given by

$$\beta_{t+1}^i = \beta_t^i - \mu \hat{H}^{-1} \nabla f(\beta_t; S_i X, S_i Y) + \eta(\beta_t^i - \beta_{t-1}^i),$$

which can be expanded as

$$\beta_{t+1}^i = \beta_t^i - \mu(X^\top \hat{S}^\top \hat{S} X)^{-1}(S_i X)^\top(S_i X \beta_t^i - S_i Y) + \eta(\beta_t^i - \beta_{t-1}^i).$$

by subtracting the solution $\tilde{\beta}^i$ of the $i$-th sketched LS subproblem, we obtain

$$\beta_{t+1}^i - \tilde{\beta}^i = \beta_t^i - \tilde{\beta}^i - \mu(X^\top \hat{S}^\top \hat{S} X)^{-1}(S_i X)^\top(S_i X \beta_t^i - S_i Y) + \eta(\beta_t^i - \tilde{\beta}^i) - \eta(\beta_{t-1}^i - \tilde{\beta}^i).$$

Since the above iteration is a two-step scheme, we adopt the analysis framework of the heavy ball method (Recht, 2010), which suggest to consider the following bipartite relation

$$\begin{bmatrix} \beta_{t+1}^i - \tilde{\beta}^i \\ \beta_t^i - \tilde{\beta}^i \end{bmatrix} = \begin{bmatrix} (1 + \eta)I_d - \mu(X^\top \hat{S}^\top \hat{S} X)^{-1}(S_i X)^\top S_i X & -\eta I_d \\ I_d & 0 \end{bmatrix} \begin{bmatrix} \beta_t^i - \tilde{\beta}^i \\ \beta_{t-1}^i - \tilde{\beta}^i \end{bmatrix}.$$

The above relation leads to

$$\begin{bmatrix} D_X V_X^\top (\beta_{t+1}^i - \tilde{\beta}^i) \\ D_X V_X^\top (\beta_t^i - \tilde{\beta}^i) \end{bmatrix} = \begin{bmatrix} W_i + \eta I_d & -\eta I_d \\ I_d & 0 \end{bmatrix} \begin{bmatrix} D_X V_X^\top (\beta_t^i - \tilde{\beta}^i) \\ D_X V_X^\top (\beta_{t-1}^i - \tilde{\beta}^i) \end{bmatrix}. \tag{13}$$

where $W_i = I_d - \mu (U_X^\top \hat{S}^\top \hat{S} U_X)^{-1} U_X^\top S_i^\top S_i U_X$. Now, for all $i$, we consider the spectral properties of the iteration matrix

$$L^{(i)} \triangleq \begin{bmatrix} W_i + \eta I_d & -\eta I_d \\ I_d & 0 \end{bmatrix}.$$

Let the eigenvalue decomposition of $W_i$ be $W_i = \tilde{U}_i \Lambda_i \tilde{U}_i^\top$, where $\Lambda_i$ is a real diagonal matrix with its $k$-th diagonal $\lambda_k^{(i)}$ being the $k$-th eigenvalue of $W_i$. Together with the following permutation $\Pi$, with its entries defined as

$$\Pi_{i,j} = \begin{cases} 1 & \text{if } i \text{ is odd and } j = i, \\ 1 & \text{if } i \text{ is even and } j = d + i, \\ 0 & \text{otherwise,} \end{cases}$$

we can construct a similar transformation

$$P = \begin{bmatrix} \tilde{U}_i & 0 \\ 0 & \tilde{U}_i \end{bmatrix} \Pi,$$

which leads to a factorization of the iteration matrix

$$L^{(i)} = P^{-1} \begin{bmatrix} L_1^{(i)} & 0 & \cdots & 0 \\ 0 & L_2^{(i)} & \cdots & 0 \\ \vdots & \ddots & \ddots & \vdots \\ 0 & 0 & \cdots & L_d^{(i)} \end{bmatrix} P.$$

For $k = 1, \ldots, d$, the blocks $L_k^{(i)}$ can be expressed as

$$L_k^{(i)} = \begin{bmatrix} \eta + \lambda_k^{(i)} & -\eta \\ 1 & 0 \end{bmatrix},$$

whose characteristic polynomial is of the form

$$u^2 - (\eta + \lambda_k^{(i)}) u + \eta = 0.$$

Based on the condition

$$(\eta + \lambda_k^{(i)})^2 \leq 4\eta, \tag{14}$$

both eigenvalues of $L_k^{(i)}$ are imaginary and have a magnitude of $\sqrt{\eta}$. To ensure this condition holds for all $\lambda_k^{(i)}$, we must determine the appropriate value of $\eta$.

Due to the conditions $0 < \epsilon < 1/10$ and $|\mu - 1| \leq 1/4$, it follows from Lemma A.2 and Lemma A.5 that, with probability at least $1 - 2\delta$, we have $\rho(W_i) \leq \frac{(\mu+1)\epsilon + |\mu-1|}{1-\epsilon} \leq 19/36$. Consequently, for a fixed $k$, the eigenvalues satisfy $|\lambda_k^{(i)}| \leq \rho(W_i) \leq 19/36$.

To minimize the contraction ratio $\sqrt{\eta}$, we solve the inequality equation 14 based on the bound of $\rho(W_i)$, yielding $53/36 - \sqrt{17}/3 \leq \eta \leq 53/36 + \sqrt{17}/3$ (refer to Remark A.6). Then we have the minimum $\eta_{\min} = 53/36 - \sqrt{17}/3 < 1/9$, which leads to $\rho(L^{(i)}) = \sqrt{\eta_{\min}} < 1/3$

By repeatedly applying the relation (13), we have

$$\begin{bmatrix} D_X V_X^\top (\beta_{a_i}^i - \tilde{\beta}^i) \\ D_X V_X^\top (\beta_{a_i-1}^i - \tilde{\beta}^i) \end{bmatrix} = \begin{bmatrix} W_i + \eta I_d & -\eta I_d \\ I_d & 0 \end{bmatrix}^{a_i} \begin{bmatrix} D_X V_X^\top (\beta_0^i - \tilde{\beta}^i) \\ D_X V_X^\top (\beta_{-1}^i - \tilde{\beta}^i) \end{bmatrix},$$

which leads to

$$\left\| \begin{bmatrix} D_X V_X^\top (\beta_{a_i}^i - \tilde{\beta}^i) \\ D_X V_X^\top (\beta_{a_i-1}^i - \tilde{\beta}^i) \end{bmatrix} \right\| \leq \left\| [L^{(i)}]^{a_i} \right\| \left\| \begin{bmatrix} D_X V_X^\top (\beta_0^i - \tilde{\beta}^i) \\ D_X V_X^\top (\beta_{-1}^i - \tilde{\beta}^i) \end{bmatrix} \right\|.$$

Thanks to the relation $X = U_X D_X V_X^\top$ and the column orthogonality of $U_X$, the above inequality is equivalent to

$$\left\| \begin{bmatrix} X(\beta_{a_i}^i - \tilde{\beta}^i) \\ X(\beta_{a_i-1}^i - \tilde{\beta}^i) \end{bmatrix} \right\| \le \|[L^{(i)}]^{a_i}\| \left\| \begin{bmatrix} X(\beta_0^i - \tilde{\beta}^i) \\ X(\beta_{-1}^i - \tilde{\beta}^i) \end{bmatrix} \right\|,$$

and squaring the above inequality yields

$$\|X(\beta_{a_i}^i - \tilde{\beta}^i)\|^2 + \|X(\beta_{a_i-1}^i - \tilde{\beta}^i)\|^2 \le \|[L^{(i)}]^{a_i}\|^2 \left( \|X(\beta_0^i - \tilde{\beta}^i)\|^2 + \|X(\beta_{-1}^i - \tilde{\beta}^i)\|^2 \right).$$

Together with $\beta_{-1}^i = \beta_0^i = \beta_{a_{i-1}}^{i-1}$, we obtain

$$\|X(\beta_{a_i}^i - \tilde{\beta}^i)\|^2 \le \|[L^{(i)}]^{a_i}\sqrt{2}\|^2 \|X(\beta_0^i - \tilde{\beta}^i)\|^2.$$

By taking advantage of the Gelfand Formula (Kozyakin, 2009), we have

$$\lim_{a_i \to +\infty} \|[L^{(i)}]^{a_i}\sqrt{2}\|^{\frac{1}{a_i}} = \lim_{a_i \to +\infty} \|[L^{(i)}]^{a_i}\|^{\frac{1}{a_i}} (\sqrt{2})^{\frac{1}{a_i}}$$

$$= \lim_{a_i \to +\infty} \|[L^{(i)}]^{a_i}\|^{\frac{1}{a_i}}$$

$$= \rho(L^{(i)}).$$

Therefore, for any $\tau > 0$ satisfying $\rho(L^{(i)}) + \tau < \frac{1}{3}$, there exists a constant $M_i > 0$ such that if $a_i > M_i$,

$$\|[L^{(i)}]^{a_i}\sqrt{2}\|^{\frac{1}{a_i}} - \rho(L^{(i)}) < \tau,$$

or equivalently,

$$\|[L^{(i)}]^{a_i}\sqrt{2}\| < [\rho(L^{(i)}) + \tau]^{a_i}$$

$$< (\frac{1}{3})^{a_i}.$$

Then, it follows that

$$\|X(\beta_{a_i}^i - \tilde{\beta}^i)\| \le (\frac{1}{3})^{a_i} \|X(\beta_0^i - \tilde{\beta}^i)\|. \tag{15}$$

To investigate the relationship between the true parameter vector $\beta$ and the iterate $\beta_{a_i}^i$, we consider the following inequality

$$\|X(\beta_{a_i}^i - \beta)\| \le \|X(\beta_{a_i}^i - \tilde{\beta}^i)\| + \|X(\tilde{\beta}^i - \beta)\|$$

$$\le (\frac{1}{3})^{a_i} \|X(\beta_0^i - \tilde{\beta}^i)\| + \|X(\tilde{\beta}^i - \beta)\|$$

$$\le (\frac{1}{3})^{a_i} \|X(\beta_0^i - \beta)\| + \left[1 + (\frac{1}{3})^{a_i}\right] \|X(\tilde{\beta}^i - \beta)\|, \tag{16}$$

by taking expectation over the random noise $\zeta$ on both sides, we have

$$\mathbb{E}\|X(\beta_{a_i}^i - \beta)\| \le (\frac{1}{3})^{a_i} \mathbb{E}\left\|X(\beta_0^i - \beta)\right\| + \left[1 + (\frac{1}{3})^{a_i}\right] \mathbb{E}\|X(\tilde{\beta}^i - \beta)\|$$

$\square$

*Remark* A.6. Based on the condition $\rho(W_i) \le \frac{(\mu+1)\epsilon+|\mu-1|}{1-\epsilon} \le Z$ with $Z$ being a positive constant, the inequality equation 14 leads to the following bounds

$$(2 - Z) - 2\sqrt{1 - Z} \le \eta \le (2 - Z) + 2\sqrt{1 - Z}.$$

## A.3 PROOF OF THEOREM 4.2

Before presenting the proof of this theorem, we first state some lemmas and remarks.

Lemma A.7 is a reformulation of Theorem 2.4 from (Dobriban & Liu, 2019). We use it to explore the loss of accuracy of the $i$-th sketched LS estimator against the OLS estimator when using *Sketch-and-Solve* methods with SRHT sketching matrix of different sizes.

**Lemma A.7.** *For the $i$-th sketched LS subproblem, let $S$ be an $N \times N$ subsampled randomized Hadamard matrix. Suppose also that $X$ is an $N \times d$ deterministic matrix whose empirical spectral distribution converges weakly to some fixed probability distribution with compact support bounded away from the origin. Then as $N$ tends to infinity, while $d/N \to \gamma \in (0,1)$, $m_i/N \to \xi \in (\gamma, 1)$, the $i$-th relative prediction efficiency $\mathrm{PE}(i)$ has the limit*

$$\mathrm{PE}(i) = \frac{\mathbb{E}\|X\tilde{\beta}^i - X\beta\|^2}{\mathbb{E}\|X\hat{\beta} - X\beta\|^2} = \frac{1 - \gamma}{\xi - \gamma}.$$

*Remark* A.8. Let $g(\gamma, \xi) = (1-\gamma)/(\xi - \gamma)$, and it is monotonically increasing with respect to $\gamma$ and monotonically decreasing with respect to $\xi$. According to Lemma A.7, when $\xi > m_i/N > d/N > \gamma$, it follows that

$$\mathrm{PE}(i) \leq \frac{1 - d/N}{m_i/N - d/N}$$
$$= \frac{N - d}{m_i - d},$$

or equivalently,

$$\mathbb{E}\|X\tilde{\beta}^i - X\beta\|^2 \leq \frac{N - d}{m_i - d}\mathbb{E}\|X\hat{\beta} - X\beta\|^2.$$

Furthermore, if the condition below holds

$$\mathrm{Var}(\|X\tilde{\beta}^i - X\beta\|) \geq \mathrm{Var}(\sqrt{\frac{N - d}{m_i - d}}\|X\hat{\beta} - X\beta\|),$$

we obtain

$$\mathbb{E}\|X\tilde{\beta}^i - X\beta\| \leq \sqrt{\frac{N - d}{m_i - d}}\mathbb{E}\|X\hat{\beta} - X\beta\|.$$

Therefore, the subsequent discussions are based on the following conditions:

- $\xi > m_i/N > d/N > \gamma$;

- $\mathrm{Var}(\|X\tilde{\beta}^i - X\beta\|) \geq \mathrm{Var}(\sqrt{(N-d)/(m_i - d)}\|X\hat{\beta} - X\beta\|).$

**Lemma A.9.** *In Algorithm 2, let $N, m_1$ be powers of 2, $\delta \in (0,1)$, $\epsilon \in (0, 1/10)$, $|\mu - 1| \leq 1/4$ and $\eta = 53/36 - \sqrt{17}/3$. Let*

$$r \geq c\epsilon^{-2}\left[d + \log\left(\frac{N}{\delta}\right)\right]\log\left(\frac{ed}{\delta}\right), \quad m_1 > r,$$

*where $c > 0$ is a constant and $T^\dagger = \sum_{i=1}^{K} a_i$. There exists a constant $M^\dagger > 0$, such that if $a_i > M^\dagger$, $i = 1, \ldots, K$, with probability at least $1 - 2\delta$, for any $\beta_0 \in \mathbb{R}^d$, it holds that*

$$\mathbb{E}\|X(\beta_{T^\dagger} - \beta)\| \leq (\frac{1}{3})^{T^\dagger}\mathbb{E}\|X(\beta_0 - \beta)\| + \left[\sqrt{\frac{N - d}{m_K - d}} + o(1)\right]\mathbb{E}\|X(\hat{\beta} - \beta)\|,$$

*and $o(1)$ is an infinitesimal as $M^\dagger \to +\infty$.*

***Proof.*** In SLSE-FRS, we construct $K$ sketched LS subproblems and utilize the M-IHS method to take iterations in each sketched LS subproblem.

We begin with the 1st sketched LS subproblem with an initial guess $\beta_0 \in \mathbb{R}^d$, and we set the initial iterates $\beta_{-1}^1 = \beta_0^1 = \beta_0$. According to Theorem 4.1 and Lemma A.7, there exists a constant $M_1 > 0$, such that if $a_1 > M_1$, it holds that

$$\mathbb{E}\|X(\beta_{a_1}^1 - \beta)\| \leq (\frac{1}{3})^{a_1}\mathbb{E}\|X(\beta_0^1 - \beta)\| + \left[1 + (\frac{1}{3})^{a_1}\right]\sqrt{\frac{N - d}{m_1 - d}}\mathbb{E}\|X(\hat{\beta} - \beta)\|. \qquad (17)$$

In the subsequent skecthed LS subproblems, we take the following initial iterates, i.e., for $i = 2, \ldots, K$,

$$\beta_{-1}^i = \beta_0^i = \beta_{a_{i-1}}^{i-1}.$$

In the 2nd sketched LS subproblem, there exists a constant $M_2 > 0$, such that if $a_2 > M_2$, we have

$$\mathbb{E}\|X(\beta_{a_2}^2 - \beta)\| \leq (\frac{1}{3})^{a_2}\mathbb{E}\|X(\beta_0^2 - \beta)\| + \left[1 + (\frac{1}{3})^{a_2}\right]\sqrt{\frac{N-d}{m_2-d}}\mathbb{E}\|X(\hat{\beta} - \beta)\|$$

$$\leq (\frac{1}{3})^{a_2}\mathbb{E}\|X(\beta_{a_1}^1 - \beta)\| + \left[1 + (\frac{1}{3})^{a_2}\right]\sqrt{\frac{N-d}{m_2-d}}\mathbb{E}\|X(\hat{\beta} - \beta)\|$$

$$\leq (\frac{1}{3})^{a_1+a_2}\mathbb{E}\|X(\beta_0 - \beta)\| + \left\{\left[1 + (\frac{1}{3})^{a_2}\right]\sqrt{\frac{N-d}{m_2-d}}\right.$$

$$\left. + (\frac{1}{3})^{a_2}\left[1 + (\frac{1}{3})^{a_1}\right]\sqrt{\frac{N-d}{m_1-d}}\right\}\mathbb{E}\|X(\hat{\beta} - \beta)\|, \tag{18}$$

where the 3rd inequality follows from inequality (17).

Then, by deduction, there exists $K$ constants $\{M_i\}$, $i = 1, \ldots, K$, such that if $a_i > M_i$, we obtain following relation

$$\mathbb{E}\|X(\beta_{a_K}^K - \beta)\| \leq \left[\prod_{i=1}^K (\frac{1}{3})^{a_i}\right]\mathbb{E}\|X(\beta_0 - \beta)\| + \sqrt{\frac{N-d}{m_K-d}}\mathbb{E}\|X(\hat{\beta} - \beta)\| \tag{19}$$

$$+ \left\{\sum_{i=1}^{K-1}\left[1 + (\frac{1}{3})^{a_i}\right]\sqrt{\frac{N-d}{m_i-d}}\prod_{j=i+1}^K (\frac{1}{3})^{a_j}\right\}\mathbb{E}\|X(\hat{\beta} - \beta)\|.$$

In addition, when $M_K$ tends to $+\infty$, it holds that

$$\left\{\sum_{i=1}^{K-1}\left[1 + (\frac{1}{3})^{a_i}\right]\sqrt{\frac{N-d}{m_i-d}}\prod_{j=i+1}^K (\frac{1}{3})^{a_j}\right\} = o(1), \tag{20}$$

and the quantity (20) decays to 0 at an exponential rate.

Let $M^\dagger = \max\{M_i \mid i = 1, \ldots, K\}$, if $a_i > M^\dagger$, $i = 1, \ldots, K$, together with (20), the relation (19) can be further simplified as follows

$$\mathbb{E}\|X(\beta_{T^\dagger} - \beta)\| \leq (\frac{1}{3})^{T^\dagger}\mathbb{E}\|X(\beta_0 - \beta)\| + \left[\sqrt{\frac{N-d}{m_K-d}} + o(1)\right]\mathbb{E}\|X(\hat{\beta} - \beta)\|.$$

which is the claimed conclusion. $\qquad\square$

The following lemma derives the convergence behavior of the 2nd stage of SLSE-FRS.

**Lemma A.10.** *Suppose that the conditions of Therorem 4.2 hold, there exists a constant $M^* > 0$, such that if $T > T^\dagger + M^*$, it follows that*

$$\mathbb{E}\|X(\beta_T - \hat{\beta})\| \leq (\frac{1}{3})^{T-T^\dagger}\mathbb{E}\|X(\beta_{T^\dagger} - \hat{\beta})\|.$$

***Proof.*** After $T^\dagger$ iterations at the 1st stage of SLSE-FRS, for $t = T^\dagger + 1, \ldots, T$, we employ M-IHS to solve the original full-scale LS problem. The update formula is given by

$$\beta_{t+1} = \beta_t - \mu\hat{H}^{-1}\nabla f(\beta_t; X, Y) + \eta(\beta_t - \beta_{t-1})$$

with $\hat{H} = X^\top\hat{S}^\top\hat{S}X$. Subtracting the exact OLS estimator $\hat{\beta}$, we obtain

$$\beta_{t+1} - \hat{\beta} = \beta_t - \hat{\beta} - \mu(X^\top\hat{S}^\top\hat{S}X)^{-1}X^\top(X\beta_t - Y) + \eta(\beta_t - \hat{\beta}) - \eta(\beta_{t-1} - \hat{\beta}).$$

Similar to the proof of Theorem 4.1, we consider the following bipartite relation

$$\begin{bmatrix} \beta_{t+1} - \hat{\beta} \\ \beta_t - \hat{\beta} \end{bmatrix} = \begin{bmatrix} (1+\eta)I_d - \mu(X^\top \hat{S}^\top \hat{S}X)^{-1}X^\top X & -\eta I_d \\ I_d & 0 \end{bmatrix} \begin{bmatrix} \beta_t - \hat{\beta} \\ \beta_{t-1} - \hat{\beta} \end{bmatrix}.$$

By multiplying $D_X V_X^\top$ from left on both sides of the above relation, it leads to

$$\begin{bmatrix} D_X V_X^\top(\beta_{t+1} - \hat{\beta}) \\ D_X V_X^\top(\beta_t - \hat{\beta}) \end{bmatrix} = \begin{bmatrix} W + \eta I_d & -\eta I_d \\ I_d & 0 \end{bmatrix} \begin{bmatrix} D_X V_X^\top(\beta_t - \hat{\beta}) \\ D_X V_X^\top(\beta_{t-1} - \hat{\beta}) \end{bmatrix},$$

where $W = I_d - \mu(U_X^\top \hat{S}^\top \hat{S} U_X)^{-1}$. Now, we consider the spectral properties of the iteration matrix

$$L \triangleq \begin{bmatrix} W + \eta I_d & -\eta I_d \\ I_d & 0 \end{bmatrix},$$

and a similar strategy adopted in the proof of Theorem 4.1 (i.e., constructing a transformation matrix $\hat{P}$) leads to the following factorization

$$L = \hat{P}^{-1} \begin{bmatrix} L_1 & 0 & \cdots & 0 \\ 0 & L_2 & \cdots & 0 \\ \vdots & \ddots & \ddots & \vdots \\ 0 & 0 & \cdots & L_n \end{bmatrix} \hat{P},$$

where each block $L_k$ is given by

$$L_k = \begin{bmatrix} \eta + \hat{\lambda}_k & -\eta_1 \\ 1 & 0 \end{bmatrix},$$

and $\hat{\lambda}_k$ is the $k$-th eigenvalue of $W$. The characteristic polynomial of $L_k$ is of the form

$$u^2 - (\eta + \hat{\lambda}_k)u + \eta = 0.$$

Based on the condition

$$(\eta + \hat{\lambda}_k)^2 \leq 4\eta, \tag{21}$$

both of the eigenvalues of $L_k$ are imaginary and have a magnitude of $\sqrt{\eta}$. To ensure the above condition holds for all $\hat{\lambda}_k$, we have to determine the value of $\eta$.

Since $0 < \epsilon < 1/10$ and $|\mu - 1| \leq 1/4$, it follows from Lemma A.2, A.4, and A.5 that, with probability at least $1 - 2\delta$, the spectral radius $\rho(W)$ is bounded as follows

$$\rho(W) \leq \frac{\epsilon + |\mu - 1|}{1 - \epsilon} \leq \frac{(\mu + 1)\epsilon + |\mu - 1|}{1 - \epsilon} \leq \frac{19}{36}.$$

Consequently, all the eigenvalues $\hat{\lambda}_k$ of $W$ are bounded, i.e.,

$$|\hat{\lambda}_k| \leq \rho(W) \leq \frac{19}{36}.$$

By adopting a similar argument as in the proof of Theorem 4.1, we obtain that, there exists a constant $M^\star > 0$, such that if $t - T^\dagger > M^\star$, it holds that

$$\|X(\beta_t - \hat{\beta})\| \leq (\frac{1}{3})^t \|X(\beta_{T^\dagger} - \hat{\beta})\|.$$

Then, if $T - T^\dagger > M^\star$, we have

$$\mathbb{E}\|X(\beta_T - \hat{\beta})\| \leq (\frac{1}{3})^{T - T^\dagger} \mathbb{E}\|X(\beta_{T^\dagger} - \hat{\beta})\|$$

$\square$

**Proof of Theorem 4.2.** If $T - T^\dagger > M^*$, due to Lemma A.10, we have

$$\mathbb{E}\|X(\beta_T - \hat{\beta})\| \leq (\frac{1}{3})^{T - T^\dagger} \mathbb{E}\|X(\beta_{T^\dagger} - \hat{\beta})\|.$$

Then it follows that

$$\mathbb{E}\|X(\beta_T - \beta)\| \leq \mathbb{E}\|X(\beta_T - \hat{\beta})\| + \mathbb{E}\|X(\hat{\beta} - \beta)\|$$

$$\leq (\frac{1}{3})^{T-T^{\dagger}}\mathbb{E}\|X(\beta_{T^{\dagger}} - \hat{\beta})\| + \mathbb{E}\|X(\hat{\beta} - \beta)\|$$

If $\min\{T - T^{\dagger}, a_i\} > M^*$, for $i = 1, \ldots, K$, it holds that

$$\mathbb{E}\|X(\beta_T - \beta)\| \leq (\frac{1}{3})^T\mathbb{E}\|X(\beta_0 - \beta)\| + \mathbb{E}\|X(\hat{\beta} - \beta)\|$$

$$+ (\frac{1}{3})^{T-T^{\dagger}}\left[\sqrt{\frac{N-d}{m_K - d}} + o(1)\right]\mathbb{E}\|X(\hat{\beta} - \beta)\|.$$

Moreover, there exists a constant $M^{\circ} > 0$, such that if $T - T^{\dagger} > M^{\circ}$ and $M^{\circ}$ tends to $+\infty$, it reads that

$$(\frac{1}{3})^{T-T^{\dagger}}\left[\sqrt{\frac{N-d}{m_K - d}} + o(1)\right] = o(1),$$

and the above quantity decays exponentially. Let $M = \max(M^*, M^{\circ})$, if $a_i > M$, $i = 1, \ldots, K$, and $T > T^{\dagger} + M$, then the conclusion follows. □

### A.4 LOWER BOUND OF $a_1$

The following theorem provides an lower bound of the iteration count $a_1$ for the 1st sketched LS subproblem.

**Theorem A.11.** *In SLSE-FRS, given a prescribed precision $\omega \in (0, 1)$, the number of iterations $a_1$ needed for the 1st sketched LS subproblem to full fill (12) satisfies*

$$a_1 \geq \log_3\left(\frac{\mathbb{E}\|X(\beta_0 - \tilde{\beta}_1)\|}{\omega\mathbb{E}\|X(\tilde{\beta}_1 - \beta)\|}\right).$$

*Proof.* To achieve the prescribed precision $\omega$, i.e., (12) being full filled, the iteration terminates when the following condition is met, i.e.,

$$\mathbb{E}\|X(\beta_{a_1}^1 - \tilde{\beta}^1)\| + \mathbb{E}\|X(\tilde{\beta}^1 - \beta)\| \leq (1 + \omega)\mathbb{E}\|X(\tilde{\beta}^1 - \beta)\|.$$

In fact, we need

$$\mathbb{E}\|X(\beta_{a_1}^1 - \tilde{\beta}^1)\| \leq \omega\mathbb{E}\|X(\tilde{\beta}^1 - \beta)\|.$$

According to (15) in the proof of Theorem 4.1, it holds that

$$\mathbb{E}\|X(\beta_{a_1}^1 - \tilde{\beta}^1)\| \leq (\frac{1}{3})^{a_1}\mathbb{E}\|X(\beta_0 - \tilde{\beta}^1)\|.$$

By requiring

$$(\frac{1}{3})^{a_1}\mathbb{E}\|X(\beta_0 - \tilde{\beta}^1)\| \leq \omega\mathbb{E}\|X(\tilde{\beta}^1 - \beta)\|,$$

and taking logarithms on both sides of the above inequality, then the claimed result is obtained. □

### A.5 PROOF OF THEOREM 4.3

In this subsection, we provide the proof of the lower bound of $a_i$ for the $i$-th sketched LS subproblem.

*Proof.* In the $i$-th sketched LS subproblem, to achieve the specified precision $\omega$, i.e., (12) being satisfied, the iteration stops when the following condition meets

$$\mathbb{E}\|X(\beta_{a_i}^i - \tilde{\beta}^i)\| + \mathbb{E}\|X(\tilde{\beta}^i - \beta)\| \leq (1 + \omega)\mathbb{E}\|X(\tilde{\beta}^i - \beta)\|.$$

In fact, we need

$$\mathbb{E}\|X(\beta_{a_i}^i - \tilde{\beta}^i)\| \leq \omega\mathbb{E}\|X(\tilde{\beta}^i - \beta)\|. \tag{22}$$

According to (15) in the proof of Theorem 4.1, it follows that

$$\mathbb{E}\|X(\beta_{a_i}^i - \tilde{\beta}^i)\| \leq (\frac{1}{3})^{a_i}\mathbb{E}\|X(\beta_0^i - \tilde{\beta}^i)\|.$$

Based on the relation between the initial iterate $\beta_0^i = \beta_{a_{i-1}}^{i-1}$ and the exact solution of the $i$-th sketched LS subproblem, we have

$$\mathbb{E}\|X(\beta_{a_{i-1}}^{i-1} - \tilde{\beta}^i)\| \leq \mathbb{E}\|X(\beta_{a_{i-1}}^{i-1} - \tilde{\beta}^{i-1})\| + \mathbb{E}\|X(\tilde{\beta}^{i-1} - \beta)\| + \mathbb{E}\|X(\tilde{\beta}^i - \beta)\|. \quad (23)$$

In a similar fashion of Lemma A.7 and Remark A.8, as $N$ tends to infinity, we let $d/N \to \gamma \in (0,1)$, $m_i/N \to \xi_i \in (\gamma,1)$. When the following condition holds

$$\xi_i > \frac{m_i}{N} > \frac{d}{N} > \gamma,$$

we obtain

$$r(i-1,i) = \sqrt{\frac{m_i - d}{m_{i-1} - d}} \geq \frac{\mathbb{E}\|X(\tilde{\beta}^{i-1} - \beta)\|}{\mathbb{E}\|X(\tilde{\beta}^i - \beta)\|},$$

which leads to

$$\mathbb{E}\|X(\tilde{\beta}^{i-1} - \beta)\| \leq r(i-1,i)\mathbb{E}\|X(\tilde{\beta}^i - \beta)\|. \quad (24)$$

Since the iteration of the $(i-1)$-th sketched LS subproblem also stops when the precision $\omega$ is achieved, we get

$$\mathbb{E}\|X(\beta_{a_{i-1}}^{i-1} - \tilde{\beta}^{i-1})\| \leq \omega\mathbb{E}\|X(\tilde{\beta}^{i-1} - \beta)\|. \quad (25)$$

By substituting (24) and (25) into (23), it follows that

$$\mathbb{E}\|X(\beta_{a_{i-1}}^{i-1} - \tilde{\beta}^i)\| \leq [(1+\omega)r(i-1,i)+1]\mathbb{E}\|X(\tilde{\beta}^i - \beta)\| \quad (26)$$

The inequality (22) holds if

$$(\frac{1}{3})^{a_i}[(1+\omega)r(i-1,i)+1]\mathbb{E}\|X(\tilde{\beta}^i - \beta)\| \leq \omega\mathbb{E}\|X(\tilde{\beta}^i - \beta)\|.$$

By taking logarithms on both sides of the above inequality, it results in

$$a_i \geq \log_3\left[\frac{(1+\omega)r(i-1,i)+1}{\omega}\right].$$

Since $\omega \in (0,1)$ and $[(1+\omega)r(i-1,i)+1] > 1$, it reads that $a_i > 0$. $\qquad\square$

### A.6 PROOF OF THEOREM 4.4

The proof of this theorem below is given in the asymptotic sense, that is, in the sense of $N \to +\infty$.

***Proof.*** The conditions $m_{i+1}/m_i = 2$ and $m_K = N/2$ lead to $m_i = 2^{-(K+1-i)}N$, together with $d/N \to \gamma \in (0, 2^{-K})$ as $N \to +\infty$, it follows that $r(i-1,i) = \sqrt{(1 - 2^{K+2-i}\gamma)/(2 - 2^{K+2-i}\gamma)}$ ($i = 2,\dots,K$). Since $2^{K+2-i}\gamma < 1$, the function $r(x) = \sqrt{(1-x)/(2-x)}$ monotonically decreases with respect to $x \in (0,1)$, and the function $\ell(r) = \log_3\{[1 + (1+\omega)r]/\omega\}$ monotonically increases with respect to $r > 0$, it holds that $a_i = \log_3\{[1 + (1+\omega)r(i-1,i)]/\omega\}$ monotonically increases with repect to $i = 2,\dots,K$, i.e., $a_2 < \dots < a_K$. Due to the above results, we have

$$a_2 = \log_3\{[1 + (1+\omega)\sqrt{(1 - 2^K\gamma)/(2 - 2^K\gamma)}]/\omega\}$$
$$> \log_3(1/\omega)$$

and

$$a_K = \log_3\{[1 + (1+\omega)\sqrt{(1 - 2^2\gamma)/(2 - 2^2\gamma)}]/\omega\}$$
$$< \log_3\{[1 + (1+\omega)/\sqrt{2}]/\omega\} = \alpha,$$

which lead to the fact $\log_3(1/\omega) < a_i < \alpha$ ($i = 2,\dots,K$). For $i = 1$, according to Appendix A.4, taking the lower bound $a_1 = \log_3[(1/\omega)(\mathbb{E}\|X(\beta_0 - \tilde{\beta}^1)\|/\mathbb{E}\|X(\tilde{\beta}^1 - \beta)\|)]$, together with the

condition $1 < \mathbb{E}\|X(\beta_0 - \tilde{\beta}_1)\|/\mathbb{E}\|X(\tilde{\beta}_1 - \beta)\| < 1 + (1 + \omega)/\sqrt{2}$, it follows that $\log_3(1/\omega) < a_1 < \alpha$.

Now, we consider the dominant costs of all stages of Algorithm 2. At the beginning, the main workload in the initialization stage includes the construction of all of the required data, that is, $(S_0 X, S_0 Y)$, $(S_i X, S_i Y)$ $(i = 1, \ldots, K)$, and $\hat{H}$. According to the discussion in Section 3, the cost of this stage is dominated by applying the Hadamard transform to a matrix of size $N \times d$, which is dominated by $Nd \log_2 N$ FLOPS.

Next, we will consider the two stages of iteration. Since Algorithm 2 terminates when a noise level estimator is attained, which means that the 1st term of the upper bound in (11) has decreased to $\sigma$-level. Due to the fact that the initial error $\mathbb{E}\|X(\beta_0 - \beta)\|$ is a constant, it is equivalent to require the factor $(1/3)^T$ reduces to $\sigma$-level, i.e., $(1/3)^T \leq \sigma$. Therefore, the total count $T$ of iteration is bounded below by $T \geq \log_3(1/\sigma)$. To minimize the cost of Algorithm 2 in this discussion, we let $T = \log_3(1/\sigma)$.

In the 1st stage of iteration, Algorithm 2 takes $a_i$ iterations in the $i$-th sketched LS subproblem, the total $K$ sketched subproblems requires $\sum_{i=1}^K a_i[(4d+1)m_i + 2d^2 + 5d]$ FLOPs. Thanks to the fact $\sum_{i=1}^K m_i = (1 - 2^{-K})N$, together with the upper bound $a_i < \alpha$, it holds that the dominant cost in this stage is bounded by $4\alpha Nd$.

In the 2nd stage of iteration, Algorithm 2 takes full-scale iterations, and the iteration count reads

$$
\begin{aligned}
T - T^\dagger &= \log_3(1/\sigma) - \sum_{i=1}^K a_i \\
&< \log_3(1/\sigma) - K\log_3(1/\omega) = \log_3(\omega^K/\sigma).
\end{aligned}
$$

The main cost of each full-scale iteration comes from the computation of gradients $\nabla f(\beta_t; X, Y)$, which is dominated by $4Nd$. Together with the above bound of $T - T^\dagger$, it follows that the dominant cost in this stage is bounded by $4[\log_3(\omega^K/\sigma)]Nd > 0$ with $\omega > \sigma^{1/K}$. $\qquad \square$

### A.7 DETERMINATION OF $a_i$

In this subsection, we aim to determine feasible values of $a_i$ for all subsequent experiments at a sufficiently low cost to ensure the efficiency of SLSE-FRS.

According to Theorem 4.3, let $a_i$ be its lower bound. We test SLSE-FRS on an LS problem with $N = 2^{20}$, $d = 2^6$, and $\kappa = 10^4$. When $\omega$ takes values from the set $\{2^{-4}, 2^{-3}, 2^{-2}, 2^{-1}, 1\}$, we obtain the curve on the left plot of Figure 3, , where the $\omega$-axis uses a logarithmic scale. Since $a_i$ is monotonically increasing with respect to $i = 2, \ldots, K$, we only present the curves for $a_2$ and $a_K$, with other curves necessarily lying between them. The two curves almost coincide, with the curve of $a_K$ slightly higher than that of $a_2$, and both curves decrease linearly as $\log(\omega)$ increases. As $\omega$ characterizes the solution precision of all sketched LS subproblems, the curves in this plot indicate that as the solution precision improves, the value of $a_i$ increases. Among all tested values of $\omega$, the maximum value of $a_i$ is approximately 3. This suggests that the value of $a_i$ does not need to be too large, if high-precision solutions (related to small $\omega$) for sketched LS subproblems are not required.

Based on the aforementioned facts, we conducted further experiments. Specifically, we set $a_i = \text{NoI}$ (the number of iterations of the sketched LS subproblems) for all $i$. The right plot in Figure 3 shows the curve of the computing time of SLSE-FRS versus the value of NoI. This curve indicates that the total computing time of SLSE-FRS achieves its minimum at $\text{NoI} = 2$, while its total computing time at $\text{NoI} = 3$ is very close to that at $\text{NoI} = 2$. Based on these observations, we recommend using $a_i = 2$ or 3 for all $i$ in subsequent experiments.

### A.8 LARGER SCALE EXPERIMENT

We provide one larger-scale experiment with $N = 2^{20}$, $d = 2^{10}$, $\kappa = 10^8$ in Figure 4. We only compare IDS with SLSE-FRS, excluding PCG and M-IHS due to their much longer computing time. In this setting, SLSE-FRS still outperforms IDS. We believe that SLSE-FRS will maintain its superior performance on even larger-scale problems.

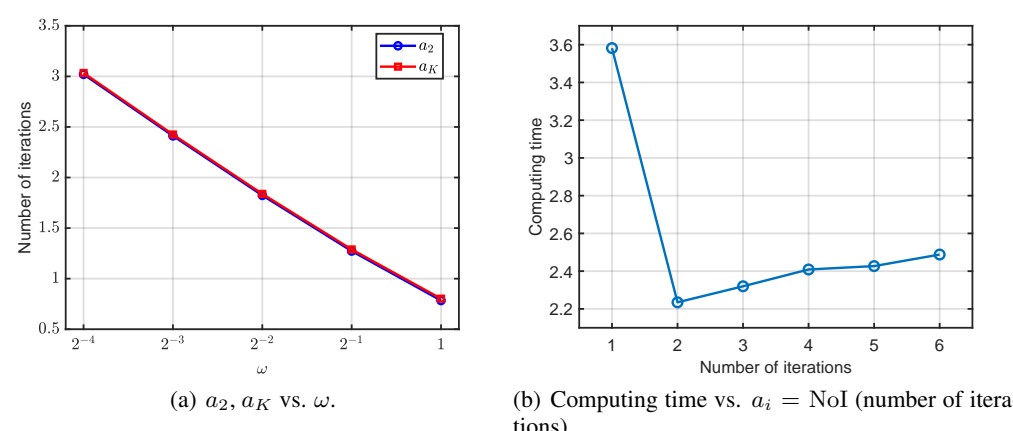

(a) $a_2$, $a_K$ vs. $\omega$.

(b) Computing time vs. $a_i = $ NoI (number of iterations).

Figure 3: The iteration control parameter $a_i$ for SLSE-FRS: $a_2$, $a_K$ versus $\omega$; computing time versus $a_i = $ NoI (the number of iterations of the sketched LS subproblems) for all $i$.

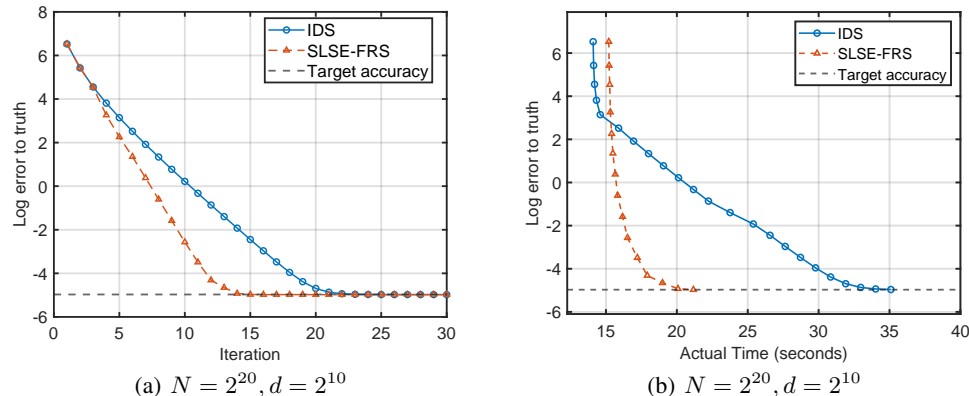

(a) $N = 2^{20}, d = 2^{10}$

(b) $N = 2^{20}, d = 2^{10}$

Figure 4: $\Delta_t$ versus iterations and actual computing time for SLSE-FRS and IDS.

### A.9 SKETCH SIZE & ITERATION STEP TUNING

In this experiment, we consider the possibility of further improving the computational efficiency of SLSE-FRS by tuning its sketch size sequence $m_i$ and iteration step in large and challenging LS problems. We set the sample size $N = 2^{22}$, the feature size $d = 2^6$, and the condition number $\kappa = 10^8$. If we continue to increase the sketch size by a factor of 2, we will inevitably end up solving several large-scale sketched LS subproblems (e.g., $m_i = 2^{20}$ or $2^{21}$), which remains computationally expensive. Therefore, we opted for the **tuning technique** as a more flexible approach. We only construct a few large-scale sketched LS subproblems. In this test, SLSE-FRS-tuning only constructs sketched LS subproblems of small sketch sizes $\{2^{10}, 2^{11}, 2^{12}, 2^{13}, 2^{14}, 2^{15}, 2^{16}\}$ and one larger sketch size $\{2^{19}\}$. We set $a_i = 3$ for each sketched LS subproblem.

In Figure 5, we demonstrate the performance of SLSE-FRS, IDS, PCG and SLSE-FRS-tuning. We observe that SLSE-FRS maintains a significant lead in both convergence rate and computational efficiency, thereby improving the state-of-the-art performance of high-precision LS estimators. Combining with the tuning technique, SLSE-FRS-tuning can further reduce the computing time.

We have only found a sequence of sketch sizes that improved the computational efficiency of SLSE-FRS in this experiment, but we currently do not have a good strategy to automatically tune this sequence. The experimental results indicate that the tuning technique has the potential to enhance the computational efficiency of SLSE-FRS, which deserves further research in the future.

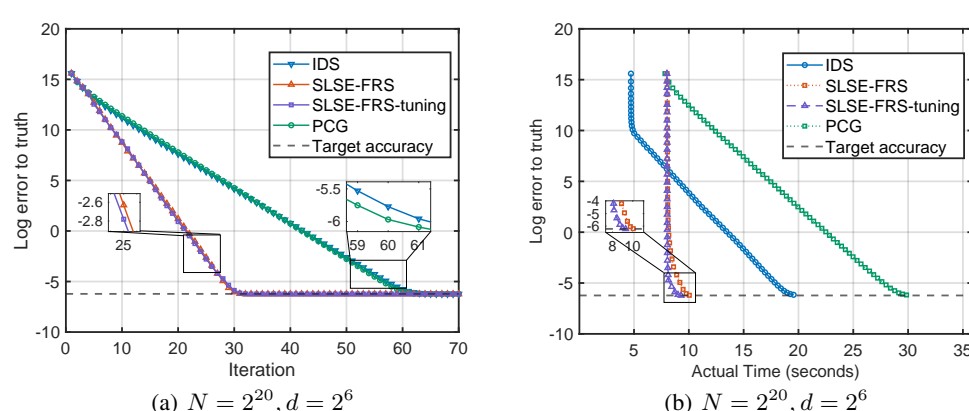

Figure 5: $\Delta_t$ versus iterations and actual computing time for SLSE-FRS and M-IHS.

## A.10 IMPLEMENTATION DETAILS

According to (Epperly, 2024), the inverse of the sketched Hessian matrix applied to a vector is implemented based on the QR factorization. Due to the inefficiency of the MATLAB function fwht, we adopt a C++-FWHT library whenever the application of the Hadamard transform is needed in the implementation of SLSE-FRS, IDS, and PCG, which can be found at `https://github.com/jeffeverett/hadamard-transform`. No other additional libraries were used in our numerical experiments.

## A.11 THE USE OF LARGE LANGUAGE MODELS (LLMs)

The LLMs were used only for language polishing. They were not involved in research design, analysis, or results. The authors take full responsibility for the content.

