# OpenReview forum: "Sequential Least-Squares Estimators with Fast Randomized Sketching for Linear Statistical Models"
_ICLR.cc/2026/Conference — Submitted to ICLR 2026_

### Official Review · Reviewer_dNyU · 2025-10-29

**Soundness:** 4
**Presentation:** 3
**Contribution:** 3
**Rating:** 6
**Confidence:** 4

**Summary:**

This paper presents Sequential Least-Squares Estimators with Fast Randomized Sketching (SLSE-FRS) for large-scale linear statistical models. The method appears to be the first to unify Sketch-and-Solve with Iterative-Sketching. It enlarges the sketch size progressively and solves a sequence of sketched least-squares subproblems, which incrementally refines the estimate of the true parameter vector. The authors also provide a systematic treatment of convergence behavior and computational complexity.

**Strengths:**

* The main contribution is the novel integration of Sketch-and-Solve and Iterative-Sketching within a single framework.
* The analysis is comprehensive:
   1)  it explains how to construct the sequence of sketched subproblems to balance estimation accuracy with computational cost;
   2) it proposes a theoretically sound and computationally tractable stopping criterion to achieve optimal iterative accuracy;
  3) it establishes, both in theory and in experiments, that the estimator attains noise-level accuracy comparable to ordinary least squares (OLS).

**Weaknesses:**

1) The *o(1)* term in equation (11) of Theorem 4.2 is not quantified. Moreover, letting $M \to \infty$ appears at odds with the paper’s non-asymptotic stance. In line with IDS-style results, please specify the scale of $M$ (e.g., explicit bounds or rates). Without such quantification, the result is not fully convincing.

2) Aside from the added momentum component, the proposed algorithm is essentially equivalent to the two-stage process used in IDS. Could the authors explicitly and theoretically demonstrate how much improvement their method achieves over IDS?

3) Theorem 4.4 provides a complexity analysis but does not present an explicit convergence rate. As a result, it is not immediately clear how the proposed algorithm achieves the claimed convergence–complexity trade-off.

**Questions:**

1) It is recommended that the authors cite the following works for completeness:
Derezinski, Michal, et al. “Newton-LESS: Sparsification without trade-offs for the sketched Newton update.” Advances in Neural Information Processing Systems, 34 (2021): 2835–2847.
Garg, Sachin, Kevin Tan, and Michał Dereziński. “Distributed least squares in small space via sketching and bias reduction.” Advances in Neural Information Processing Systems, 37 (2024): 73745–73782.
These works emphasize that bias removal can significantly improve estimation accuracy and convergence performance.

2) The paper compares the proposed method with M-IHS and shows that the second-stage procedure improves performance over M-IHS. However, it lacks an ablation comparison between SLSE-FRS and the estimator (\boldsymbol{\beta}_T) obtained using only the first-stage iterative form (7). It is recommended to include such an ablation study to better isolate and demonstrate the contribution of the second stage.

3) Could the authors clarify the purpose of the condition number  $ \kappa $ used in the experiments? Specifically, how is it defined and what role does it play in the analysis or performance evaluation?

4) The paper presents numerical experiments showing that the proposed SLSE-FRS method outperforms IDS in terms of the convergence–complexity trade-off. However, it lacks an explicit, formula-based comparison of the computational complexities of SLSE-FRS and IDS. Providing such a comparison would make the claimed improvement more transparent and convincing.

---

> ### Author Response · Authors · 2025-11-23
>
> We thank the reviewer for the careful review and constructive feedback. Our detailed responses are provided below.
>
> Concern 1:
>
> The $o(1)$ term in equation (11) of Theorem 4.2 is not quantified. Moreover, letting $M\rightarrow\infty$ appears at odds with the paper’s non-asymptotic stance. In line with IDS-style results, please specify the scale of $M$ (e.g., explicit bounds or rates). Without such quantification, the result is not fully convincing.
>
> Response 1:
>
> According to equation (20), we have
> $$
> \hat{o}(1)
> <
> \left(\frac{1}{3}\right)^{M^\dagger}
> \left[
> \frac{4}{3}\sqrt{\frac{N-d}{m_1-d}}
> \right],
> $$
> where $M^\dagger$ is a non-negative integer defined in our paper.  From the last inequality in the proof of Theorem 4.2 (page 18), we have
>
> $$
> o(1)=(\frac{1}{3})^{T-T^{\dagger}}\left[\sqrt{\frac{N-d}{m_K-d}}+\hat{o}(1)\right]<(\frac{1}{3})^M\left(\frac{13}{9}\sqrt{\frac{N-d}{m_1-d}}\right).
> $$
>
> Therefore, the second term in the error bound of Theorem 4.2 satisfies
>
> $$
> (1+o(1))\mathbb{E}\||X(\hat{\beta}-\beta)\||
> <
> \left[ 1 + \left(\frac{1}{3}\right)^M \left(\frac{13}{9}\sqrt{\frac{N-d}{m_1-d}}\right) \right]
> \mathbb{E}\||X(\hat{\beta}-\beta)\||.
> $$
>
> With only a small $M$, this can be simplified to
>
> $$
> [1+o(1)]\mathbb{E}\||X(\hat{\beta}-\beta)\||<2\mathbb{E}\||X(\hat{\beta}-\beta)\||.
> $$
>
> Thus this term is at most twice the noise level (**non-asymptotic**). For larger $M$, the factor $1+o(1)$ is close to 1, so the term is essentially at the noise level. Theorem 4.2 therefore shows that once the first term of the error bound is reduced to the noise level, we already obtain an acceptable estimator of $\beta$.
>
> Regarding the constants such as $M$: Algorithm 2 is constructed based on solving each LS subproblem by M-IHS. Since M-IHS is a linear stationary iterative method, its convergence rate is determined by the spectral radius of its iteration matrix $L_i$. Because Algorithm 2 switches between different LS subproblems, we use the spectral norm $\||L_i\||$ instead of the spectral radius $\rho(L_i)$ to obtain a uniform convergence bound. The relationship between spectral norm and spectral radius follows from Gelfand's formula, which introduces constants such as $M$. These constants arise only because we provide a non-asymptotic convergence bound for a sequence of changing subproblems.
>
> Concern 2:
>
> Aside from the added momentum component, the proposed algorithm is essentially equivalent to the two-stage process used in IDS. Could the authors explicitly and theoretically demonstrate how much improvement their method achieves over IDS(formula-based comparison of the computational complexities of SLSE-FRS and IDS)?
>
> Response 2:
>
> Firstly, SLSE is a novel algorithmic framework. In fact, SLSE is inherently recursive and can be viewed either as a recursive initialization framework or as a recursive “small-sketch-warmstart + big-data polish” paradigm; see our responses to Reviewer cy5G’s Concerns 2 and 3 for a detailed discussion.
>
> Secondly, under the SLSE framework, new algorithmic instances can be constructed by combining different sketching strategies with various LS solvers, and each such instance merits further theoretical and experimental investigation.
>
> Finally, the LS subproblem sizes can be regarded as  hyperparameters within the SLSE framework. Although this issue  lies beyond the scope of the current manuscript, a systematic study aimed at identifying well-chosen hyperparameters has the potential to significantly improve the efficiency of the resulting algorithms.
>
> Regarding the practical performance of SLSE, we construct an algorithmic instance under the SLSE framework based on the M-IHS method, namely Algorithm 2. Our experiments show that Algorithm 2 clearly outperforms IDS, and we believe that even more efficient algorithms can be developed within the SLSE framework. According to the description in the IDS paper, IDS essentially uses a sketched gradient within the IHS method, whereas our SLSE framework is more general and capable of encompassing a much broader class of new algorithms.
>
> Regarding the computational complexity, compared to the IDS method, the overall computational cost of Algorithm 2 is significantly lower. Assuming Algorithm 2 takes $t$ iterations with respect to the $i$-th sketched LS subproblem, which requires $\big((4d+1)m_i + 2d^2 + 5d\big)t$ FLOPs, while $t$ iterations of IDS requires $(4d+1)\sum_{k = i}^{i + t} m_k + 2d^2 t + 5dt$ FLOPs. The leading term of the cost is reduced from $(4d+1)\sum_{k = i}^{i + t} m_k$ to $(4d+1)t m_i$, where $\{m_k\}$ is an exponentially growing sequence. This is due to Algorithm 2 taking a reasonable number of iterations to fully exploit the information from each sketched subproblem, thereby resulting in fewer full-scale iterations, which incurs a lower computational cost. In addition, Algorithm 2 (convergence rate bound $1/3$) has a faster convergence speed compared to IDS (convergence rate bound $1/2$). Therefore, the cost of Algorithm 2 is lower.

---

> ### Author Response · Authors · 2025-11-23
>
> Concern 3:
>
> Theorem 4.4 provides a complexity analysis but does not present an explicit convergence rate. As a result, it is not immediately clear how the proposed algorithm achieves the claimed convergence–complexity trade-off.
>
> Response 3:
>
> We would like to kindly point out that the convergence rates are explicitly analyzed in Theorems 4.1 and 4.2. Specifically, Theorem 4.1 provides the convergence rate bound for the sketched LS subproblems, while Theorem 4.2 establishes the convergence rate bound of the entire iterative process. In both theorems, we show that the convergence rate bound is $1 / 3$.
>
> Regarding the result in Theorem 4.4, it provides the cost of each stage of Algorithm 2, rather than the cost of a single iteration. In other words, the total cost of the stages given in Theorem 4.4 is the overall cost required by Algorithm 2 to obtain a $\sigma$ -accurate solution (noise level precision solution).
>
> Concern 4:
>
> The paper compares the proposed method with M-IHS and shows that the second-stage procedure improves performance over M-IHS. However, it lacks an ablation comparison between SLSE-FRS and the estimator $\beta_T$ obtained using only the first-stage iterative form (7). It is recommended to include such an ablation study to better isolate and demonstrate the contribution of the second stage.
>
> Response 4:
>
> This is a very good suggestion. If there were no page limitations, we could certainly perform an ablation comparison between Algorithm 2  and the estimator $\beta_{T^{\dagger}}$ obtained using only the first-stage iterative form (7).
>
> In fact, when the noise level is large, the LS estimator $\hat{\beta}$ provides a poor approximation to the true parameter $\beta$, and Algorithm 2 can achieve a solution with noise-level accuracy without entering the second stage of Algorithm 2. However, when the noise level is small (e.g., $10^{-8}$), if Algorithm 2 only performs the first-stage iterations, the error will stagnate and cannot be further reduced. Only after Algorithm 2 enters the second stage does the solution accuracy continue to decrease and reach the noise level $10^{-8}$.
>
> We briefly illustrate the contributions of the two stages of Algorithm 2 using existing experiments. The inflection point in the "error-time" plot (see the 2nd plot of Col. 3 in Fig. 1) indicates the transition between the two stages. The ratio of the number of iterations between Stage 1 and Stage 2 is approximately 3:1, while the ratio of their computing time is approximately 1:2. This shows that full-scale iterations are much more expensive. Therefore, to improve the efficiency of the algorithm, the number of iterations in Stage 2 should be reduced as much as possible.
>
> Concern 5:
>
> Could the authors clarify the purpose of the condition number used in the experiments? Specifically, how is it defined and what role does it play in the analysis or performance evaluation?
>
> Response 5:
>
> The feature matrix $X$ is generated as follows:
>
> Firstly, a Gaussian random matrix $G \in \mathbb{R}^{N \times d}$ is constructed, and its singular value decomposition is computed as
> $$
> G = U \Sigma V^{\top};
> $$
> Secondly, the singular values in $\Sigma$ are then replaced by a prescribed sequence $\{\tilde{\sigma}_i\}, i = 1,\ldots,d$ , yielding a new diagonal matrix $\tilde{\Sigma}$. Thirdly, the new feature matrix is formed as
>
> $$
> X = U \tilde{\Sigma} V^{\top}.
> $$
>
> which ensures that $X$ has the desired condition number. The sketching operation is performed directly to $X$ without column normalization.
>
> In our experiments, we set a relatively large condition number in order to fully demonstrate the performance of each algorithm under challenging scenarios. If a smaller condition number were used, the LS problem would be much easier to solve, and all algorithms would converge in few iterations, making it difficult to meaningfully compare their true performance. A large condition number increases the difficulty of the LS problem, forcing all algorithms to perform more iterations and thereby revealing their numerical behavior in many-iteration regimes. For example, if the condition number were very small, Algorithm 2 might converge without even entering its second stage.
>
> **We will incorporate the suggested references into the revised manuscript where appropriate. We sincerely thank the reviewer again for the careful consideration, valuable suggestions, and constructive feedback, which have greatly helped improve the clarity and quality of our work.**

---

### Official Review · Reviewer_cy5G · 2025-10-29

**Soundness:** 3
**Presentation:** 3
**Contribution:** 2
**Rating:** 4
**Confidence:** 3

**Summary:**

This paper introduces a two-stage framework that addresses a sequence of increasingly large sketched least-squares subproblems, followed by a few full LS iterations to achieve OLS-level accuracy. The authors provide a contraction guarantee (under specific SRHT-based conditions and momentum selection) and an implementation using M-IHS. Experiments demonstrate faster convergence than traditional random sketched least square methods.

**Strengths:**

$\textbf{Clear two-stage recipe.}$ Stage-1 builds $K$ sketched LS problems with growing sketch sizes and uses those solutions to warm-start Stage-2 on full data; the framework is solver-agnostic and explains the cost savings intuitively.

 $\textbf{Convergence statement and practical parameters.}$ Theorems provide an exponential decay bound and a global contraction rate for suitable $(\mu,\eta)$, with a data-driven surrogate stopping rule (via $\omega$) and a lower bound on the per-subproblem iteration count $a_i$.

$\textbf{Empirical speedups.}$ Plots/tables indicate faster time-to-accuracy than IDS and PCG on synthetic settings; M-IHS comparisons are also included.

**Weaknesses:**

$\textbf{Scope and assumptions feel narrow.}$ The method is largely a combination of standard components—Sketch-and-Solve least squares, iterative Hessian sketching (IHS), SRHT/CountSketch embeddings, and preconditioned Richardson/gradient iterations. The contribution appears only to lie in how these blocks are combined.

$\textbf{Literature position.}$ The authors need to clarify whether similar “sketch-warmstart + full-data polish” patterns have ever been explored, and articulate the specific differences with this literature.

$\textbf{Ablations.}$ Due to the combination nature of the paper, I expect more ablation studies to be performed to determine which components are the key to the success of the performance. See questions.

$\textbf{Unclear synthetic settings.}$ Experiments have some unclear and unexplained settings, see questions..

**Questions:**

1. To better evaluate the paper's “integration-and-scheduling” perspective is reasonable, but novelty claims should be framed accordingly. Clear positioning is recommended by clearly stating that each component is established.

2. Why do the simulation settings have unrealistically small noise level $\sigma^2=1e-8$? This narrows the problem, only placing the problem in an ultra–high-SNR regime. Adding a noise sweep is necessary.

3. The synthetic data description is under-specified: while $X$ is said to be i.i.d. Gaussian with an “artificially adjusted” condition number and
$\beta$ is Gaussian, the paper does not describe the actual procedure to impose a target condition number $\kappa$ on $X$ (e.g., singular-value planting vs. diagonal scaling), nor whether columns are standardized, whether an intercept is used, or how
$X$ is normalized before sketching.

4. Additional ablations are necessary. e.g., are there differences between one-shot big sketch vs. the schedule? How about the comparison between fixed estimated Hessian vs. updating after several iterations?

---

> ### Author Response · Authors · 2025-11-23
>
> We thank the reviewer for the detailed and constructive feedback on our manuscript. We have carefully considered all comments and provide our responses below.
>
> Concern 1:
>
> The method is largely a combination of standard components—Sketch-and-Solve least squares, iterative Hessian sketching (IHS), SRHT/CountSketch embeddings, and preconditioned Richardson/gradient iterations. The contribution appears only to lie in how these blocks are combined.
>
> Response 1:
>
> Our **main contribution** is to propose the new SLSE framework (**recursive initialization framework**), and the construction of a **efficient instance of SLSE based on the M-IHS method**, namely Algorithm 2. Based on Algorithm 2, the effectiveness of the SLSE framework has been preliminarily validated, and therefore it is necessary to conduct further research on this framework. For a detailed discussion of the novelty of the SLSE framework, please refer to the following response.
>
> Concern 2:
>
> The authors need to clarify whether similar “sketch-warmstart + full-data polish” patterns have ever been explored, and articulate the specific differences with this literature.
>
> Response 2:
>
> When proposing the SLSE framework, we did not recognize that it could be interpreted through the perspective of the “sketch-warmstart + full-data polish” paradigm, which we agree is a highly insightful perspective. Our initial motivation was to unify Sketch-and-Solve with Iterative-Sketching. Fundamentally, SLSE can be regarded as a recursive initialization framework: each LS subproblem uses the output of the previous subproblem as its initialization, and this recursive structure ensures that the computational cost of each iteration remains low.
>
> **Following your** suggested “sketch-warmstart + full-data polish” **viewpoint**, we further propose **a new interpretation of SLSE as a recursive “small-sketch-warmstart + big-data polish” paradigm**: each subproblem serves as a “small-sketch warmstart” for the next subproblem, while simultaneously acting as a “big-data polish” for the previous one.
>
> Concern 3:
>
> Due to the combination nature of the paper, I expect more ablation studies to be performed to determine which components are the key to the success of the performance. To better evaluate the paper's “integration-and-scheduling” perspective is reasonable, but novelty claims should be framed accordingly. Clear positioning is recommended by clearly stating that each component is established.
>
> Response 3:
>
> Firstly, SLSE is a novel algorithmic framework. As discussed above, SLSE is inherently recursive in nature and can be viewed either as a recursive initialization framework or as a recursive small-sketch-warmstart + big-data polish paradigm.
>
> Secondly, under the SLSE framework, new algorithmic instances can be constructed by combining different sketching techniques with various LS solvers, and each such instance merits further theoretical and experimental investigation.
>
> Finally, the LS subproblem sizes can be regarded as hyperparameters within the SLSE framework. Although this issue lies beyond the scope of the current manuscript, a more in-depth study aimed at identifying well-chosen hyperparameters has the potential to significantly improve the efficiency of the resulting algorithms.
>
> In general, ablation studies can focus on the components that constitute the SLSE framework, including the SL solvers, sketching matrices, and hyperparameters (LS subproblem sizes). Current research indicates that these components are all the key to the success of the performance of SLSE. Taking the sketching matrix as an example, CountSketch (sparse sketching matrix) is superior to SRHT in computing time, see Fig.2 (page 9).
>
> Concern 4:
>
> Why do the simulation settings have unrealistically small noise level $\sigma^2 = 10^{-8}$?
> This narrows the problem, only placing the problem in an ultra--high-SNR regime.
>
> Response 4:
>
> It should be emphasized that the applicability of the new SLSE framework and Algorithm 2 based on M-IHS is not restricted by the noise level. In our experiments, the selection of noise level is not intended to be consistent with real-world scenarios. Instead, we adopt a very small noise level in order to more clearly reveal the performance of different algorithms under challenging accuracy requirements. If large noise levels were used, the linear model approximation accuracy would be relatively low, and all algorithms would require only a few iterations to reach this accuracy, which would not be conducive to a meaningful comparison of their true performance. By contrast, a small noise level leads to a much higher target approximation accuracy, forcing all algorithms to perform more iterations in order to attain sufficient precision, thereby exposing their numerical behavior in large-iteration-count regimes. For example, if the noise level were high, Algorithm 2 might not even need to enter its second stage to obtain an approximate solution with noise-level accuracy.

---

> ### Author Response · Authors · 2025-11-23
>
> Concern 5:
>
> The synthetic data description is under-specified: while $X$ is said to be i.i.d. Gaussian with an “artificially adjusted” condition number and $\beta$ is Gaussian, the paper does not describe the actual procedure to impose a target condition number $\kappa$ on $X$ (e.g., singular-value planting vs. diagonal scaling), nor whether columns are standardized, whether an intercept is used, or how $X$ is normalized before sketching.
>
> Response 5:
>
> The feature matrix $X$ is generated as follows:
>
> Firstly, a Gaussian random matrix $G \in \mathbb{R}^{N \times d}$ is constructed, and its singular value decomposition is computed as
> $$
> G = U \Sigma V^{\top};
> $$
> Secondly, the singular values in $\Sigma$ are then replaced by a prescribed sequence $\{\tilde{\sigma}_i\}, i = 1,\ldots,d$ , yielding a new diagonal matrix $\tilde{\Sigma}$. Thirdly, the new feature matrix is formed as
>
> $$
> X = U \tilde{\Sigma} V^{\top}.
> $$
>
> which ensures that $X$ has the desired condition number. The sketching operation is performed directly to $X$ without column normalization.
>
> Concern 6:
>
> Additional ablations are necessary. e.g., are there differences between one-shot big sketch vs. the schedule? How about the comparison between fixed estimated Hessian vs. updating after several iterations?
>
> Response 6:
>
> In practice, two-stage iterative algorithms based on a one-shot large sketch are unable to match the computational efficiency of Algorithm 2.  Algorithm 2 uses a sequence of LS subproblems to construct an efficient iterative scheme. Each subproblem provides a better initialization for the next, allowing Algorithm 2 to match the convergence rate of M-IHS while significantly reducing computation time (see Col.3 in Fig.1, ). In Stage 1, Algorithm 2 solves smaller LS subproblems per-iteration compared to the one-shot big sketch approach, while maintaining nearly the same convergence rate (see the slopes of all iteration curves). The construction cost of this sequence of subproblems is dominated by applying the Hadamard transform to the original data matrix, and the cost of constructing smaller LS subproblems is negligible. Therefore, iterating over a sequence of smaller subproblems preserves a convergence rate comparable to that of the full (large-scale) problem, while significantly reducing the per-iteration computational cost. Given these advantages, it is natural to adopt this strategy.
>
> Moreover, updating the Hessian approximation inevitably increases the computational cost. In our practical experiments, however, such an updating strategy does not lead to a noticeable acceleration in the convergence rate. As a result, the additional computational overhead is not compensated by corresponding performance gains, and the updating strategy therefore offers no substantial advantage over using a fixed Hessian approximation. Similar conclusions can be found in Jonathan Lacotte and Mert Pilanci, Faster Least Squares Optimization, arXiv:1911.02675, 2021.
>
> **We sincerely thank the reviewer again for the careful reading of our rebuttal and for the constructive comments and suggestions.**

---

### Official Review · Reviewer_jZBq · 2025-10-30

**Soundness:** 3
**Presentation:** 2
**Contribution:** 2
**Rating:** 2
**Confidence:** 4

**Summary:**

The authors provide a randomized algorithm for the least squares problem. The algorithm is simple, and amounts to repeatedly performing a sketch-and-solve procedure with a doubling sketch size until some tolerance is achieved. In practice, this is done with a sketched Hessian + sketched gradient update. After which, the authors perform sketched Hessian updates on the full gradient. The authors show a high-probability guarantee for the first stage, and an asymptotic guarantee for the second. The number of iterations of the first stage is lower bounded by a logarithmic term in the reciprocal of the tolerance.

**Strengths:**

The idea of likening a series of sketch-and-solve solutions to an iterative sketched Hessian + sketched gradient method for computing an approximate least squares solution is nice and may be original. The proofs are rigorous. The writing within the paper is above average in quality. If the algorithm works well in practice, it could improve the performance of libraries for randomized methods for least squares by improving the warm-starts given to sketch-and-precondition solvers.

**Weaknesses:**

1. The algorithm is very simple, and amounts to using the doubling trick to determine the sketch size that should be employed when performing a sketch-and-solve operation. The authors solve each sketch-and-solve problem along the way efficiently via sketching the data, computing the sketched gradient, and then performing a sketched Hessian update on the coefficient obtained from the last iteration with the sketched gradient. Once the optimal sketch size is found, the authors then employ a sketched Hessian update on the full gradient.
- The problem with this is that the authors ultimately still need a sketch size $r$ on the order of $d/\epsilon^2$, as seen in Theorems 4.1 and 4.2. In a sense, we already know the necessary sketch size, or at least its order. The doubling trick allows for only a logarithmic inflation, but it is difficult to see this as anything more than an engineering tweak.
- After this, a sketched Hessian update is employed, that obtains an asymptotically linear convergence rate. I say asymptotically because the second term only goes to zero asymptotically at an unknown rate, and it is unclear how large the "constant" $M$ should be. This guarantee is not very sophisticated, and I believe the proof of this is folklore.

2. The paper does not compare the runtime of its method to other papers within the literature that solve the least squares problem with sparse sketches, e.g. Garg et al. (2024), Chenakkod et al. (2024), Anari et al. (2022), etc. One can generally expect to solve the problem (of the first stage) in $\text{nnz}(A) + O(d^2/ / \epsilon)$ time with sparse sketches. It is odd that the procedure does not yield an improvement from $O(Nd)$ to $\text{nnz}(A)$. I suspect that this is an issue with the analysis.
- In Theorem 4.2 the authors rely on a large constant $M$ and ultimately employ an asymptotic analysis of the runtime, in contrast to the non-asymptotic analyses common in the literature.
- Accordingly, the benefit of this rather complicated first stage is not evident in the guarantee for the second stage. If these guarantees are tight (I do not think they are), then there is no benefit to performing the first stage over simply initializing the sketch-and-precondition second stage with a sketch-and-solve solution.
- As such, the numerical experiments are somewhat unfair to IDS and PCG -- they are not initialized with a sketch-and-solve solution, while SLSE-FRS almost is.


### References
- Garg et al. (2024), Distributed Least Squares in Small Space via Sketching and Bias Reduction
- Chenakkod et al. (2024), Optimal Embedding Dimension for Sparse Subspace Embeddings
- Anari et al. (2022), Optimal sublinear sampling of spanning trees and determinantal point processes via average-case entropic independence.

**Questions:**

See weaknesses. This score is somewhat harsh. At the moment, I am on the fence between a 2 and a 4, and am willing to increase my score if I am proven wrong or my concerns are addressed.

---

> ### Author Response · Authors · 2025-11-20
>
> Thank you for carefully reading our paper and providing your review comments. We believe that the new SLSE framework is an innovative algorithmic framework with substantial potential for further development, and we hope to share this new idea with a broader readership. Therefore, we would be very grateful if you could read our response in full and give careful consideration to our contribution.
>
> Below, I will provide a detailed response to each of your concerns.
>
> Concern 1:
>
> The algorithm is simple, and amounts to repeatedly performing a sketch-and-solve procedure with a doubling sketch size until some tolerance is achieved. Once the optimal sketch size is found, the authors then employ a sketched Hessian update on the full gradient.
>
> Response 1:
>
> Firstly, we have to clarify that **SLSE-FRS does not aim to find an “optimal sketch size“ merely to provide a sketched Hessian for a final iteration using the full gradient**, and SLSE can be viewed as a **recursive initialization framework**, also **refer to Response 7 here and Response 2 to Reviewer cy5G**. Instead, SLSE-FRS constructs a sequence of sketched LS subproblems and performs a limited number of iterations within each subproblem (with the choice of the number of iterations supported by Theorem 4.3). The iterative solution from one subproblem is then used as the initial point for the next subproblem with a larger sketch size. Consequently, each iteration yields a progressively more accurate solution (as guaranteed by Theorems 4.1 and 4.2). Intuitively, the goal of SLSE-FRS is to obtain an estimator $\beta_T$ whose accuracy matches that of the exact LS estimator $\hat{\beta}$.
>
> We note that the exact sketched LS estimator $\tilde{\beta}^i$ of each subproblem is inherently less accurate than the exact LS estimator $\hat{\beta}$ of the original problem for the estimation of $\beta$. However, in early iterations, the current solution is far from the true parameter $\beta$, and solving a small-size sketched LS subproblem is sufficient to move the solution in the correct direction. Therefore, we fully exploit the value of each subproblem by solving it to its optimal accuracy, ensuring that
> $$
> \mathbb{E}\||X(\beta^{i} - \beta)\||
> \le (1+\omega)
> \mathbb{E}\||X(\tilde{\beta}^{i} - \beta)\||,
> $$
> This ensures that the distance between the current iterate $\beta^i$ and the true parameter $\beta$ is of the same order as the distance between the exact Sketched LS estimator $\tilde{\beta}^i$ and $\beta$ for the current sketched subproblem. Here, the tolerance $\omega$ is the accuracy requirement for the iterative solver of each LS subproblem. Moreover, all LS subproblems can share the same tolerance $\omega$, and this tolerance does not need to be very small. For example, in our experiments, performing two iterations corresponds to $\omega = 1/9$, and performing three iterations corresponds to $\omega = 1/27$.
>
> Concern 2:
>
> The authors still require a sketch size of the same order as in Theorems 4.1 and 4.2. The doubling trick thus seems to be only an engineering tweak.
>
> Response 2:
>
> The SLSE framework **does not aim to identify a single optimal sketch size**. Instead, it uses a sequence of LS subproblems to build an efficient iterative scheme. Each subproblem provides a better initialization for the next, allowing Algorithm 2 to match the convergence rate of M-IHS while significantly reducing computing time (see Col.3 in Fig.1). In Stage 1, Algorithm 2 solves only small LS subproblems per-iteration, whereas M-IHS solves the full LS problem. The sketch size increases gradually until Stage 2, where full LS problems are solved. Algorithm 2 is a M-IHS instance of the SLSE framework. The inflection point in the "error-time" plot (see the 2nd plot of Col. 3 in Fig. 1) indicates the transition between the two stages. The ratio of the number of iterations between Stage 1 and Stage 2 is approximately 3:1, while the ratio of their computing time is approximately 1:2. This shows that full-scale iterations are much more expensive. Therefore, to improve the efficiency of the algorithm, the number of iterations in Stage 2 should be reduced as much as possible.
>
> As discussed above, we only use the doubling strategy to determine the sizes of the LS subproblems of Algorithm 2. In fact, **we can treat the LS subproblem sizes as hyperparameters for extensive tuning  or learning**. Well-chosen hyperparameters can improve computational efficiency, while poorly chosen ones only affect efficiency but do not compromise the accuracy of the final approximate LS estimator, since this is guaranteed by the second stage of the SLSE framework. Please refer to Appendix A.9 for a preliminary discussion of alternative and potentially more effective strategies for selecting the sketch size beyond the doubling approach. A more in-depth study aimed at identifying good hyperparameters has the potential to significantly improve the efficiency of the resulting algorithms.

---

> ### Author Response · Authors · 2025-11-20
>
> Concern 3:
>
> The numerical experiments are somewhat unfair to IDS and PCG -- they are not initialized with a sketch-and-solve solution, while SLSE-FRS almost is.
>
> Response 3:
>
> All baseline methods are implemented exactly as specified in their original papers, and the reported runtime includes both initialization and iteration. The comparison is therefore fair. Algorithm 2 has two stages: solving a sequence of LS subproblems (Stage 1) and then solving the full LS problem (Stage 2). **Stage 1 is a recursive initialization rather than a single-step initialization**; it is an iterative process that continually produces increasingly accurate solutions while significantly reducing the per-iteration cost of M-IHS. Removing it would reduce Algorithm~2 to M-IHS. **For a more detailed discussion on the infeasibility of single-step initialization, please refer to Response 7**.
>
> Concern 4:
>
> A sketched Hessian update is employed, that obtains an asymptotically linear convergence rate. I say asymptotically because the second term only goes to zero asymptotically at an unknown rate, and it is unclear how large the "constant" should be. This guarantee is not very sophisticated, and I believe the proof of this is folklore.
>
> Response 4:
>
> According to equation (20), we have
> $$
> \hat{o}(1)
> <
> \left(\frac{1}{3}\right)^{M^\dagger}
> \left[
> \frac{4}{3}\sqrt{\frac{N-d}{m_1-d}}
> \right],
> $$
> where $M^\dagger$ is a non-negative integer defined in our manuscript.  From the last inequality in the proof of Theorem 4.2 (page 18), we have
>
> $$
> o(1)=(\frac{1}{3})^{T-T^{\dagger}}\left[\sqrt{\frac{N-d}{m_K-d}}+\hat{o}(1)\right]<(\frac{1}{3})^M\left(\frac{13}{9}\sqrt{\frac{N-d}{m_1-d}}\right).
> $$
>
> Therefore, the second term in the error bound of Theorem 4.2 satisfies
>
> $$
> [1+o(1)]\mathbb{E}\||X(\hat{\beta}-\beta)\||
> <
> \left[ 1 + \left(\frac{1}{3}\right)^M \left(\frac{13}{9}\sqrt{\frac{N-d}{m_1-d}}\right) \right]
> \mathbb{E}\||X(\hat{\beta}-\beta)\||.
> $$
>
> With only a small $M$, this can be simplified to
>
> $$
> [1+o(1)]\mathbb{E}\||X(\hat{\beta}-\beta)\||<2\mathbb{E}\||X(\hat{\beta}-\beta)\||.
> $$
>
> Thus this term is at most twice the noise level (**non-asymptotic**). For larger $M$, the factor $1+o(1)$ is close to 1, so the term is essentially at the noise level. Theorem 4.2 therefore shows that once the first term of the error bound is reduced to the noise level, we have already obtained an acceptable estimator of $\beta$.
>
> Regarding the constants such as $M$: Algorithm 2 is constructed based on solving each LS subproblem by M-IHS. Since M-IHS is a linear stationary iterative method, its convergence rate is determined by the spectral radius of its iteration matrix $L_i$. Because Algorithm 2 switches between different LS subproblems, we use the spectral norm $\||L_i\||$ instead of the spectral radius $\rho(L_i)$ to obtain a uniform convergence bound. The relationship between spectral norm and spectral radius follows from Gelfand's formula, which introduces constants such as $M$. These constants arise only because we provide a non-asymptotic convergence bound for a sequence of changing subproblems.
>
> Concern 5:
>
> The runtime is not compared with sparse-sketch methods, which can solve the first stage in $\mathrm{nnz}(A) + O(d^2 \varepsilon)$ time. The bound should improve from $O(Nd)$ to $\mathrm{nnz}(A)$.
>
> Response 5:
>
> Before providing the response, we emphasize that the quantity $\mathrm{nnz}(A)$ refers to the number of nonzeros in the matrix, and it has nothing to do with sketch methods. The runtime complexity $O(Nd)$ for Algorithm 2 is obtained under the assumption that $A$ is a dense matrix. In this case, $Nd$ is both the number of nonzeros of the dense matrix $A$ and the computational cost of multiplying $A$ by a vector. If $A$ is sparse, then the cost of multiplying a sparse matrix $A$ by a vector is approximately $\mathrm{nnz}(A)$. Accordingly, the $O(Nd)$ term in the runtime of Algorithm 2 should be replaced by $\mathrm{nnz}(A)$.
>
> Concern 6:
>
> Theorem 4.2 the authors rely on a large constant $M$ and ultimately employ an asymptotic analysis of the runtime in Theorem 4.4, in contrast to the non-asymptotic analyses common in the literature.
>
> Response 6:
>
> The asymptotic runtime analysis in Theorem 4.4 of Algorithm 2 is conducted in the regime where the dimension $N$ of the matrix $A$ is very large, rather than with respect to the constant $M$ in Theorem 4.2. Actually, in the current era of big data, it is common for the data matrix $A$ to have a very large dimension $N$.

---

> ### Author Response · Authors · 2025-11-23
>
> Concern 7:
>
> The benefit of the first stage is unclear. It seems no better than initializing the second stage with a sketch-and-solve solution.
>
> Answer 7:
>
> Based on the above discussion, Algorithm 2 under the SLSE framework is not a simple combination of an initialization step and the M-IHS method. Therefore, using a single large LS subproblem as an initialization cannot achieve the computational efficiency of Algorithm 2. For example, in our experiments the largest LS subproblem has size $m/2$. If this subproblem were used directly for initialization, one would need to compute a high-accuracy solution of it, which cannot be achieved with only 2 or 3 iterations of M-IHS. In Algorithm 2, however, the $m/2$-sized subproblem requires only 2 or 3 iterations because the previous (smaller) subproblems provide a very good initialization.
>
> Thus, SLSE can be viewed as a **recursive initialization framework**: each LS subproblem is solved using the output of the previous subproblem as its initialization, and its output serves as the initialization for the next. This recursive structure ensures that the computational cost of each iteration remains low.
>
> **In the end, we sincerely thank the reviewer for taking the time to carefully read our rebuttal and for the valuable comments and suggestions. The reviewer’s constructive feedback has been instrumental in helping us improve the quality and clarity of our manuscript.**

---

### Meta-Review · Area_Chair_jxAn · 2026-01-06

**Summary:**

The paper presents a clear two-stage sequential sketching recipe with theoretical analysis and promising synthetic speedups. The key split is whether this is a framework-level contribution (recursive "small-sketch warm-start + big-data polish") or mainly a scheduling/integration of known components (Sketch-and-Solve, IHS/M-IHS, SRHT/CountSketch). The rebuttal clarifies misunderstandings about Stage 1 and better explains some theory/experiment choices, but the submission still falls short on novelty positioning and on experimental evidence that isolates what drives the gains.

**Reviewer Concerns:**

Novelty. Unclear separation from closely related two-stage warm-start paradigms; requests for stronger related-work positioning, including recent sparse-sketch LS literature.

Theory. Concerns about the interpretability of the large constants and the absence of a crisp, manuscript-level formula comparison to IDS and an end-to-end convergence–complexity statement.

Experiments. Synthetic regime choices (very small noise, large condition number) and missing ablations (schedule vs one-shot; Stage 1-only vs full; update-frequency variants; noise sweeps), plus baseline fairness questions.

**Reviewer Scores:**

jZBq (2): likely increase given the clarified misunderstanding, but unliklely to go above 4.

cy5G (4): unchanged.

dNyU (6): unchanged.

---

### Decision · Program_Chairs · 2026-01-26

Reject